# The Glide/Gcm fate determinant controls initiation of collective cell migration by regulating Frazzled

**Tripti Gupta[1,2,3,4], Arun Kumar[1,2,3,4†], Pierre B. Cattenoz[1,2,3,4], K VijayRaghavan[5,6], Angela Giangrande[1,2,3,4]\***

[1]Department of Functional Genomics and Cancer, Institut de Génétique et de Biologie Moléculaire et Cellulaire, Illkirch, France; [2]Centre National de la Recherche Scientifique, UMR7104, Illkirch, France; [3]Institut National de la Santé et de la Recherche Médicale, U964, Illkirch, France; [4]Université de Strasbourg, Illkirch, France; [5]Department of Developmental Biology and Genetics, Tata Institute for Fundamental Research, Bangalore, India; [6]National Centre for Biological Sciences, Tata Institute for Fundamental Research, Bangalore, India

**Abstract** Collective migration is a complex process that contributes to build precise tissue and organ architecture. Several molecules implicated in cell interactions also control collective migration, but their precise role and the finely tuned expression that orchestrates this complex developmental process are poorly understood. Here, we show that the timely and threshold expression of the Netrin receptor Frazzled triggers the initiation of glia migration in the developing *Drosophila* wing. Frazzled expression is induced by the transcription factor Glide/Gcm in a dose-dependent manner. Thus, the glial determinant also regulates the efficiency of collective migration. NetrinB but not NetrinA serves as a chemoattractant and Unc5 contributes as a repellant Netrin receptor for glia migration. Our model includes strict spatial localization of a ligand, a cell autonomously acting receptor and a fate determinant that act coordinately to direct glia toward their final destination.

**\*For correspondence:** angela@ igbmc.fr

**Present address:** [†]Department of Entomology, University of California, Riverside, United States

## Introduction

Neurons and glia show mutual reliance in many functional and developmental aspects of biology. Glia migrate collectively and over long distances to establish an intricate relationship with neurons. Defective glia migration is associated with several human diseases including glial brain tumors and defective regeneration following injury in the nervous system (*Klämbt, 2009*; *Kocsis and Waxman, 2007*; *Oudega and Xu, 2006*). Hence, a thorough understanding of the molecules involved in the process of glia migration may contribute to the development of therapeutics for these pathologies. Research progress in recent years has revealed the involvement of chemotropic cues in glia migration (*von Hilchen et al., 2010*; *Chen et al., 2010*; *Spassky et al., 2002*; *Kinrade et al., 2001*; *Liu et al., 2012*). Netrins, a class of secreted laminin-related extracellular proteins, have been described as chemotropic guidance cues for axons and migrating cells during neural development (*Lai Wing Sun et al., 2011*; *Harris et al., 1996*; *Ishii et al., 1992*; *Kennedy et al., 1994*; *Mitchell et al., 1996*; *Serafini et al., 1994*). In vertebrates, Netrins secreted by the floor plate cells act differentially on the migrating oligodendrocyte precursor cells and this differential outcome is dependant upon the type of receptors expressed in the migrating cells (*Spassky et al., 2002*; *Wolf and Friedl, 2009*; *Jarjour, 2003*; *Tsai et al., 2003*). Netrins act as chemoattractants through the DCC/Frazzled family of receptors (*von Hilchen et al., 2010*; *Lai Wing Sun et al., 2011*;

*Chan et al., 1996*; *Keino-Masu et al., 1996*; *Kolodziej et al., 1996*; *Timofeev et al., 2012*) and chemorepellants through the Unc5 receptor family (*Keleman and Dickson, 2001*; *Labrador et al., 2005*). In *Drosophila*, the two Netrins (NetA and NetB [*Harris et al., 1996*; *Mitchell et al., 1996*; *Keleman and Dickson, 2001*]) and their receptor Frazzled (Fra) mediate the attraction of embryonic longitudinal glia toward the midline (*von Hilchen et al., 2010*). Despite the extensive knowledge on these ligands and receptors, the transcriptional control underlying chemoattraction and the impact of this pathway remain largely unknown. For example, what is the specific role of the receptors and how do they contribute to the different steps of collective migration, such as initiation, maintenance and arrest? Related to this issue, which transcription factors regulate the cell-specific and timely expression of the receptors? We here investigate the chemoattraction cascade that controls cell migration using the chain of glial cells across the L1 nerve in the developing *Drosophila* wing (*Aigouy et al., 2004*, *2008*; *Berzsenyi et al., 2011*; *Kumar et al., 2015*).

We show that only one of the two *Drosophila* Netrins, NetB, serves as a chemoattractant for collective glia migration. The role of Fra is to control the time of initiation of glia migration in a dosage-dependent manner, whereas Unc5 acts as the repellant receptor that controls glial arrest. Finally, we identify the transcription factor that controls the expression of Fra at the appropriate time and levels: Glial cell deficient/Glial cell missing (Glide/Gcm or Gcm, for the sake of simplicity), the fate determinant that is expressed early and transiently in the glial lineages (*Hosoya et al., 1995*; *Vincent et al., 1996*; *Jones et al., 1995*). Thus, we find that an early gene, which regulates the expression of transcription factors that execute a specific cell fate, also regulates effector genes that controls late developmental events. To our knowledge, this is the first report showing that a fate determinant directly controls collective cell migration, prompting us to revisit the role and mode of action of these types of molecules during development.

## Results

### Frazzled expression in the glia of the developing *Drosophila* wing

Fly wings are innervated by two major sensory nerves that navigate along the so-called L1 vein located at the anterior margin (L1 nerve) and along the L3 vein (L3 nerve) (*Figure 1a–d*). Glial cells originating from the sensory organ precursors (SOPs) present on the anterior margin migrate proximally, i.e. toward the central nervous system (CNS) following the axon bundle and ensheathing it throughout its length. L1 glia initiate migration at around 18 hr After Puparium Formation (hAPF), reach the level of the Costal nerve at around 22–24 hAPF and join the glial cells on the Radius by 28 hAPF. The migratory process has been accordingly subdivided into three steps: 'Initiation', 'Costa reach' and 'Complete migration' (*Figure 1a–c*).

To gain insight into the molecular pathway that triggers collective glia migration, we first examined the expression of the Fra chemoattractant receptor by using the pan glial lines *repo-Gal4 UAS-PH-GFP* (henceforth *repo*>GFP) or *gcm-Gal4 UAS-CD8-GFP* (*gcm*>GFP), which label the glial membranes. Fra is detected in glia at the time these cells begin to move as well as in the underlying axons (*Figure 1e–g''*, *Figure 1—figure supplement 1a–c, g*). The Fra protein seems to be evenly distributed along the L1 glial chain (*Figure 1—figure supplement 1d–g*). We confirmed these data using the CoinFLP technique (*Bosch et al., 2015*) to generate WT and *fra* knock down clones (KD), obtained by means of the *UAS-fra-RNAi* line, within the same wing (*Figure 1h, i*). The WT clones covering glia along the L1 nerve and surrounding cells show expression of Fra (GFP-expressing cells in *Figure 1k–k'''*), whereas Fra levels are considerably reduced in the glia and in the surrounding cells within the KD clones (RFP expressing cells in *Figure 1j–j'''*). Glia can be identified by the expression of the pan glial marker Repo (WT glia are Repo/GFP positive, whereas *fra* KD glia are Repo/RFP positive).

Thus, migrating glial cells of the peripheral nervous system (PNS) express Fra.

Next, we studied the impact of Fra on glia migration using the semiquantitative approach described by *Kumar et al.(2015)*. In short, we assessed the percentage of wings displaying complete glia migration at 28 hAPF (migratory index: MI), as an estimation of migration efficiency (*Kumar et al., 2015*). For each genotype, at least 30 wings were analyzed. We first focused on the most characterized loss-of-function (LOF) allele *fra³* (*Kolodziej et al., 1996*). As this mutation is embryonic lethal in homozygous conditions, we analyzed *fra³* heterozygous wings and found

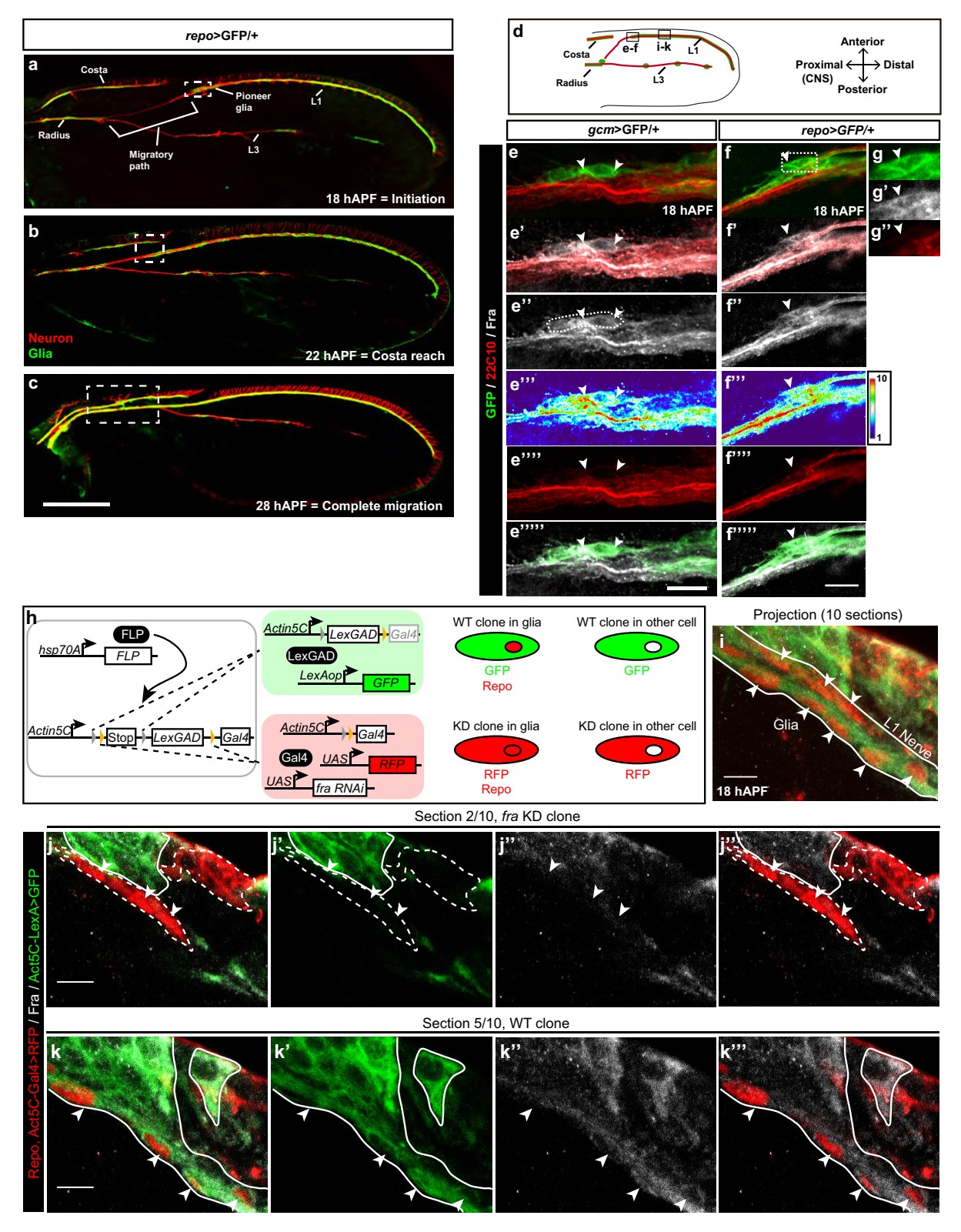

**Figure 1.** Expression of Fra in wing glia. (a–c) Immunolabeled *repo*>GFP wing (glia in green: anti-GFP; neurons in red: anti-22c10) at different stages. (a) Initiation of migration, (b) reaching the level of the costa, and (c) migration completion. (d) Schematic of a 18 hAPF developing *Drosophila* wing, insets indicate the regions shown in panels (e–f''''') and panels (i–k'''). L1 and L3 indicate L1 and L3 nerves. (e–e''''') *gcm*>GFP/+ 18 hAPF wing immunolabeled with anti-22c10 (red), anti-Fra (gray) and anti-GFP (green). mCD8-GFP was used to label the membrane. (f–g'') 18 hAPF *repo*>GFP/+

*Figure 1 continued*

wing, immunolabeled with anti-22c10 (neurons in red), anti-Fra (gray) and anti-GFP (glia in green). (e–g'') The presence of Fra in the glial soma (white arrowheads) at the front of migration. The position of the high-magnification panels (g–g'') is highlighted by the white rectangle in (f). Maximum confocal projections are shown in all figures, unless otherwise specified. White arrowheads indicate the glial cells that are expressing Fra. (h) Schematic representations of the coinFLP technique (modified from *Bosch et al. (2015)*; and the phenotypes of the different cells. (i–k''') Immunolabeled *Fra KD/ WT-coinFLP* wing at 18 hAPF. The WT clones display GFP labeling at the membranes (anti-GFP), the *fra KD* clones display RFP labeling at the membranes (anti-RFP); glial nuclei are labelled with anti-Repo in red and anti-Fra is in gray. (i) A projection of 10 confocal sections from a 18 hAPF wing. The arrowheads indicate glial cells and the white lines outline the L1 nerve. (j–k''') Individual sections: (j–j''') represents section 2/10; (k–k''') represents section 5/10, which corresponds to a deeper layer than section 2/10. (j, k) The overlay of the three channels (anti-RFP/Repo, anti-GFP and anti-Fra), (j', k') show anti-GFP alone, (j'',k'') anti-Fra and (j''', k''') the overlay of anti-RFP/Repo and anti-Fra. Glial cells are indicated by white arrowheads, the dashed lines indicate the *fra KD* clones and the continuous lines indicate the WT clones. For technical reasons, RFP (membrane labeling) and Repo (nuclear labeling) are shown in the same channel. Note the decrease in Fra levels in the *fra KD* clones. The scale bar in (a–c) represents 80 µm, in (e–f) 10 µm and in (i–k) 5 µm.

The following figure supplement is available for figure 1:

**Figure supplement 1.** Expression profile of Fra.

---

incomplete L1 glia migration in a significant fraction of samples, as shown by the position of the glial nuclei (*Figure 2a–c*). The number of glial cells is not affected and hence cannot be the cause of the migratory defect (*Figure 2d*). We reasoned that nuclei may not migrate properly, but that glial processes may still reach the final destination. To test this possibility, we assessed the migratory index of glial cells labelled by the >GFP transgene, which allowed us to visualize glial processes in flies that were heterozygous for the .. These wings also show incomplete glia migration as shown by the position of glial GFP labeling (*Figure 2e–g*) suggesting that glial cells require Fra to complete their migration.

In summary, the Fra receptor is expressed in glial cells and is seemingly necessary for their efficient migration.

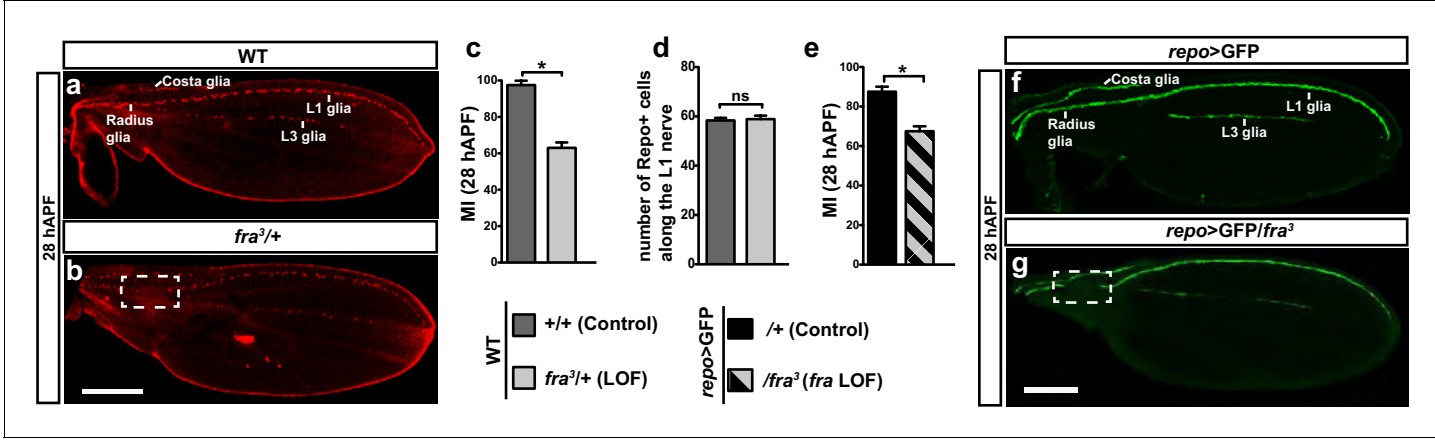

**Figure 2.** Role of Fra in wing glia. (a, b) 28 hAPF wings labeled with anti-Repo (glial nuclei): (a) WT wing showing complete migration;(b) *fra³/+* wing showing incomplete migration (dashed box). (c) Histogram representing the migratory index (MI) of the indicated genotypes, calculated using nuclear labeling (anti-Repo). The MI indicates the percentage of wings displaying complete migration (i.e. in which the glial chain reaches the proximally located glia on the Radius nerve) and was assessed at 28 hAPF unless otherwise specified. (d) Histogram representing the number of glial nuclei in the indicated genotypes. (e) Histogram representing the MI of the indicated genotypes. The MI was calculated using the membrane GFP transgenic line (*UAS*-mCD8-GFP). (f, g) 28 hAPF wings labeled with anti-GFP (glial processes): (f) *repo>GFP* wing showing complete migration; (g) *repo>GFP/fra³/+* wing showing incomplete migration (dashed box). In this and in the following figures, stars indicate p values: ***p<0.0001; **p<0.001; *p<0.05. Bars indicate the s.e.m. In this and in the following graphs on fixed wings, n≥30. Scale bars: 80 µm.

The following source data is available for figure 2:

**Source data 1.** Migratory index and repo count of of *fra³* wings in WT background.

## *fra* plays an instructive role in L1 glia migration

The lethality of *fra*[3]homozygous mutation and the expression of Fra in glia as well as in neurons prompted us to assess the role of glial *fra* expression in migration specifically. The knock down of *fra* using the *gcm*>GFP driver, which is the earliest glial driver, reveals a significant decrease in migration efficiency as compared to that observed in the control wings (*Figure 3a*, compare blank and light blue columns). To exclude the possibility of off-target effects, we analyzed wings that express the *UAS-fra-RNAi* together with the *UAS-fra* transgene, and found complete rescue of the migratory phenotype induced by the *fra* KD (*Figure 3a*, patterned light and dark blue column). This strongly suggests that the RNAi line induces a specific phenotype and that *fra* acts in a cell autonomous manner. Finally, as a complementary approach, we reintroduced *fra* expression only in the glial cells of *fra*[3]/+ animals (*Figure 3—figure supplement 1a*), and found that this rescues the migratory phenotype, albeit partially. Given the high levels of Fra expression in the *gcm*>*fra* GOF wings (*Figure 3g–g''''*), it is unlikely that this partial rescue is due to suboptimal levels of Fra. Rather, the lack of total rescue may be ascribed to the neuronal requirement of Fra. *Figure 1e–f'''''* shows that Fra is expressed in glia as well as in neurons, and indeed around one third of *fra*[3]/+ wings show an axonal navigation phenotype (*Figure 3—figure supplement 1b–d*). In these wings, axonal navigation is delayed, which may indirectly affect glial migration. To further check the role of Fra neuronal expression in glia migration, we used a driver that is expressed specifically in the L1 neurons: *nsyb-Gal4* (*West et al., 2015*; *Pauli et al., 2008*; *Riabinina et al., 2015*). *nsyb*-driven expression starts to be detected around 18 hAPF; but the expression is more prominent at 22 hAPF and onwards (*Figure 3—figure supplement 1f–h''*). We then knocked down *fra* specifically in neurons using *nsyb-Gal4* and found no delayed glia migration (*Figure 3—figure supplement 1i*), except in one third of the wings, which present an axonal navigation defect (not shown). This is in line with the above-mentioned *fra*[3]/+ data and with previous data showing that axon navigation defects indirectly affect glia migration (*Giangrande et al., 1993*).

We then asked whether Fra has an instructive role in glia migration and assessed whether migration is more efficient upon overexpressing *fra* in glia using the <u>*gcm-Gal4*</u> driver (in *fra* GOF lines). We first checked the MI of *fra* GOF wings at 28 hAPF and found that the percentage of wings that shows complete migration is higher as compared to that of control wings (*Figure 3a*, compare white and dark blue columns). Since most control wings show complete migration by 28 hAPF (90%), we also analyzed an earlier stage, when migration has been achieved in only a few control wings (24 hAPF; 12.5%). We found that many more *fra*-overexpressing wings show complete migration (68.5%, *Figure 3b*). This strongly suggests that high doses of Fra significantly increase the efficiency of glia migration. Fra levels in *fra* KD and *fra* GOF conditions are indeed reduced and increased, respectively, as compared to those observed in control wings (*Figure 3c–h*).

To clarify why the migratory efficiency decreases in *fra* KD animals, we performed time-lapse analysis and found that reducing the levels of Fra affects the first step of migration (initiation), as glia start to migrate later than the control chain (*Figure 3i*, compare the initiation step of white and light blue columns and *Figure 3—figure supplement 1e*). Accordingly, *fra*-overexpressing glia start migrating earlier than control glia, indicating that the phenotype is due to precocious initiation (*Figure 3i–k*, compare the initiation step of white and dark blue columns), and that this is associated with precocious Fra accumulation in *fra* GOF glial cells compared to that in control glia (*Figure 3l–o''*). Finally, the live imaging data demonstrates that the speed of migration at the time of initiation is not higher in *fra* GOF glia than in control glia, showing that the observed phenotype is due to precocious initiation rather than to acceleration (*Figure 3—figure supplement 2a*).

The cytoplasmic tail of Fra is known to play a major role in mediating Fra-dependent attractive responses in vivo and in cell culture studies (*Bashaw and Goodman, 1999*; *Hong et al., 1999*; *Ming et al., 1997*). We therefore asked whether this region is important in mediating glia migration. A transgenic construct that lacks the Fra intracellular cytoplasmic domain was previously described as a dominant negative mutation (*Garbe et al., 2007*) and indeed the expression of this reporter significantly reduces migration efficiency (*Figure 3a*, compare white and green-gray columns).

In summary, we find that acts in a cell autonomous manner in glial cells to mediate migration. Furthermore, our data show that that levels of Fra are critical for the initiation stage of glial cell migration and that this is mediated through the cytoplasmic domain of Fra.

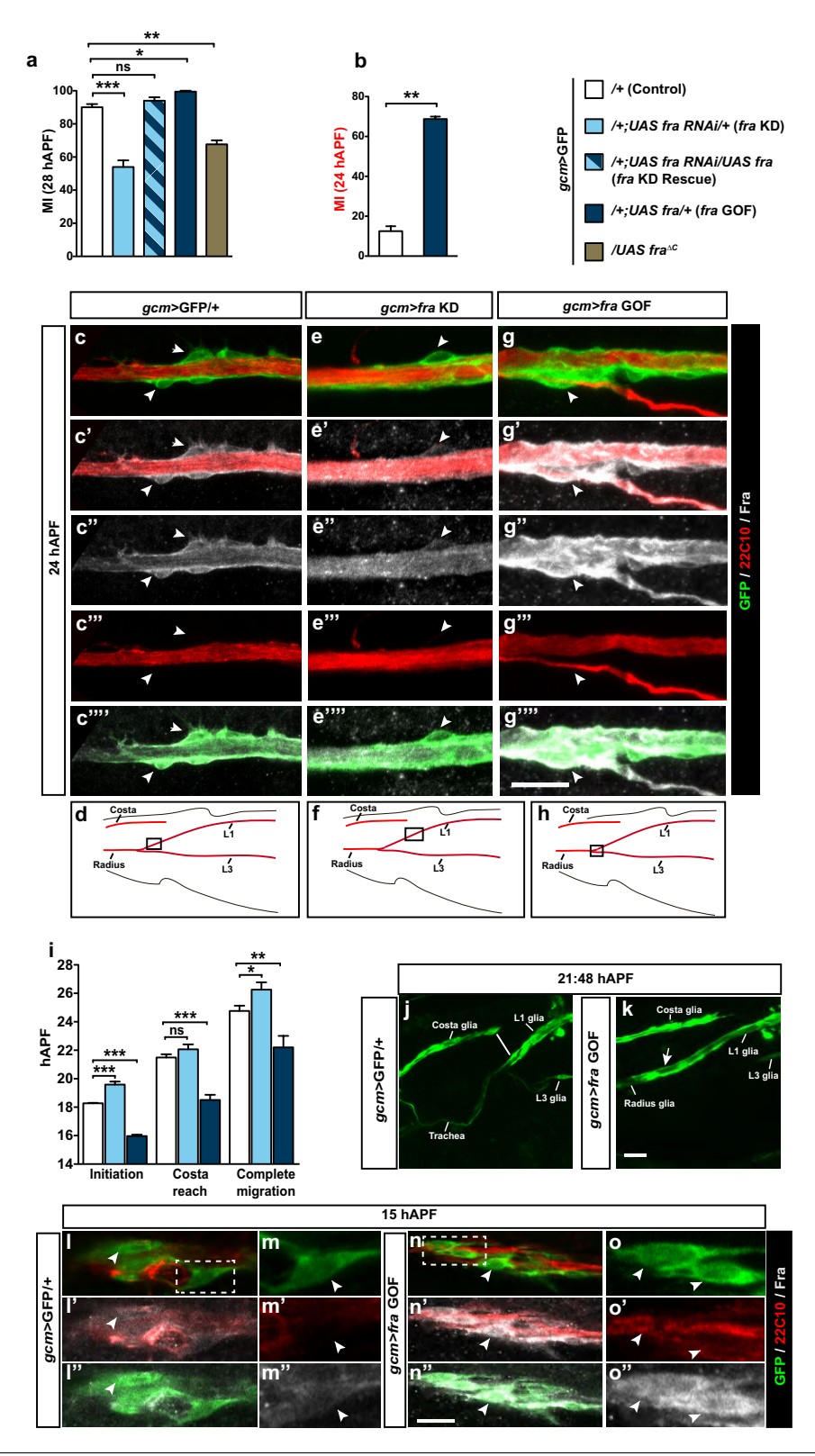

**Figure 3.** An instructive role of the chemoattractant receptor Fra in collective glia migration. (a–b) Histogram representing the MI upon *fra* knock down (*fra* KD) or overexpression (*fra* GOF) using the *gcm*>GFP/+ line. In (b), the MI was calculated at 24 hAPF. The MI was calculated using the membrane GFP transgenic line. c–c'''', e–e'''' and g–g'''' Expression profiles of Fra in *gcm*>GFP/+, *gcm*>*fra* KD and *gcm*>*fra* GOF animals at 24 hAPF. For the sake of consistency, in all genotypes, we show the glial cells that are at the front of the chain. Note the reduced protein levels in *gcm*>*fra* KD (e–e'''')

*Figure 3 continued on next page*

*Figure 3 continued*

and enhanced levels in *gcm>fra* GOF animals (**g–g′′′′**) as compared to those found in *gcm>GFP/+* animals (**c–c′′′′**). (**d, f** and **h**) Wing schematics, boxes indicate the regions shown in the above, high-magnification panels. (**i**) Graphical representation of the migratory behavior of *gcm>GFP/+*, *gcm>fra* KD and *gcm>fra* GOF wings during three highlighted phases: initiation, costa reach and complete migration (n=10). (**j, k**) Snapshots from a 21:48 hAPF time-lapse analysis of *gcm>GFP/+* and *gcm>fra* GOF wings. This corresponds to the time by which most control L1 glia (*gcm>GFP/+*) have reached the level of the Costa (white line) (**j**), whereas L1 glia overexpressing Gcm (*gcm>fra* GOF) have already completed migration by that time (white arrow) (**k**). The two panels show representative samples. (**l–o″**) Expression profiles of Fra in *gcm>GFP/+* and *gcm>fra* GOF animals at 15 hAPF. See the enhanced protein levels in *gcm>fra* GOF animals (**n–o″**) as compared to those found in *gcm>GFP/+* animals (**l–m″**). The position of the high-magnification panels (**m–m″**) is highlighted by the dashed white rectangle in (**l**), whereas that of panels (**o–o″**) is highlighted in (**n**). Please note that (**l–o″**) are comprised of few a sections rather than maximum confocal projections. Scale bars: (**c–g′′′′**), (**j–o″**), 10 μm.

The following source data and figure supplements are available for figure 3:

**Source data 1.** Migratory index and time-lapse analysis of *fra* conditional mutants in the *gcm>GFP/+* background.

**Figure supplement 1.** Migratory and neuronal defects in the *frazzled* mutation.

**Figure supplement 1—source data 1.** Summarizing the role of neurons in glia migration and the genetic interaction between *gcm* and *fra* in different transheterozygote combinations.

**Figure supplement 2.** Speed analysis and genetic interaction between *gcm* and *fra* in glia migration.

## The efficiency of glia migration depends on the dose of Gcm

Interestingly, the migratory phenotype of $fra^3/+$ glia that also carry the *gcm>GFP* driver is much stronger than that of $fra^3/+$ glia (compare *Figure 2e* patterned black an gray column, MI = 67% and *Figure 4a* patterned light gray and white column, MI = 14%). The phenotype is further enhanced in glia that express both the *gcm>GFP* and a *gcm RNAi* line (*Figure 4a* patterned brown and pink column). Since the *gcm-Gal4* driver is a hypomorphic *gcm* allele that results from the insertion of a *Gal4*-containing transposon into the *gcm* promoter (*Jacques et al., 2009*), the above result raised the possibility that Gcm and Fra interact genetically. To explore this possibility, we analyzed the glia migration phenotype in double heterozygous conditions for *fra* and two other *gcm* hypomorphic alleles, including the $gcm^{rA87}$ enhancer trap carrying a LacZ transposon inserted into the *gcm* promoter and the imprecise excision line *gcm* (*Riabinina et al., 2015*; *Vincent et al., 1996*; *Jacques et al., 2009*). In addition, we also used a *gcm* null mutation, $gcm^{N7-4}$ (*Bernardoni et al., 1997*). This confirms that reducing the dose of Gcm enhances the $fra^3$-mediated phenotype (*Figure 3—figure supplement 2b*). Furthermore, we crossed the $fra^3$ mutation with a *gcm* driver that does not affect the *gcm* locus, a transgenic line carrying 6Kb of the *gcm* promoter fused to the *Gal4* gene, which is inserted on the third chromosome (*Flici et al., 2014*). In these wings, we did not observe the enhanced migratory phenotype present in the $fra^3$, *gcm-Gal4* wings (*Figure 3—figure supplement 2c*). These findings suggest that the two genes act in the same genetic pathway that impinges on glial cell migration.

Gcm is a transiently expressed transcription factor that acts very early in glial differentiation (*Hosoya et al., 1995*; *Vincent et al., 1996*; *Jones et al., 1995*). In situ hybridization on wild-type (WT) wings has shown that the *gcm* RNA becomes detectable by 8–9 hAPF (Van de Bor and Giangrande, unpublished data) and fades in glial cells by 24 hAPF (*Popkova et al., 2012*). To clarify the role of Gcm on glia migration, we analyzed wings that are only mutant for *gcm* and used hypomorphic alleles that allow bypassing the lethality of the null mutation. Three allelic conditions were tested: $gcm$-$Gal4/gcm^{rA87}$(*gcm* LOF) (*Figure 4a*, light brown column, MI = 63%, vs. the MI = 90% of the control line *gcm>GFP/+* (white column)), *gcm-Gal4* homozygous (*Figure 4a*, orange column, MI = 41%) and $gcm^{rA87}/gcm^{N7-4}$ transheterozygous (*Figure 3—figure supplement 2b*, last column, MI = 28%) animals. Migration is indeed less efficient when the amount of Gcm is reduced and the MI is restored to normal values upon reintroduction of Gcm expression (*Figure 4a*, patterned pink columns). Finally, we used the *UAS-gcm-RNAi* line to reduce the amount of Gcm (*gcm* KD) and also observed a migratory defect (*Figure 4a*, dark brown column, MI = 25%). The rescue obtained upon co-expressing the *UAS-gcm* and the *UAS-gcm-RNAi* transgenes indicates that *gcm* plays a

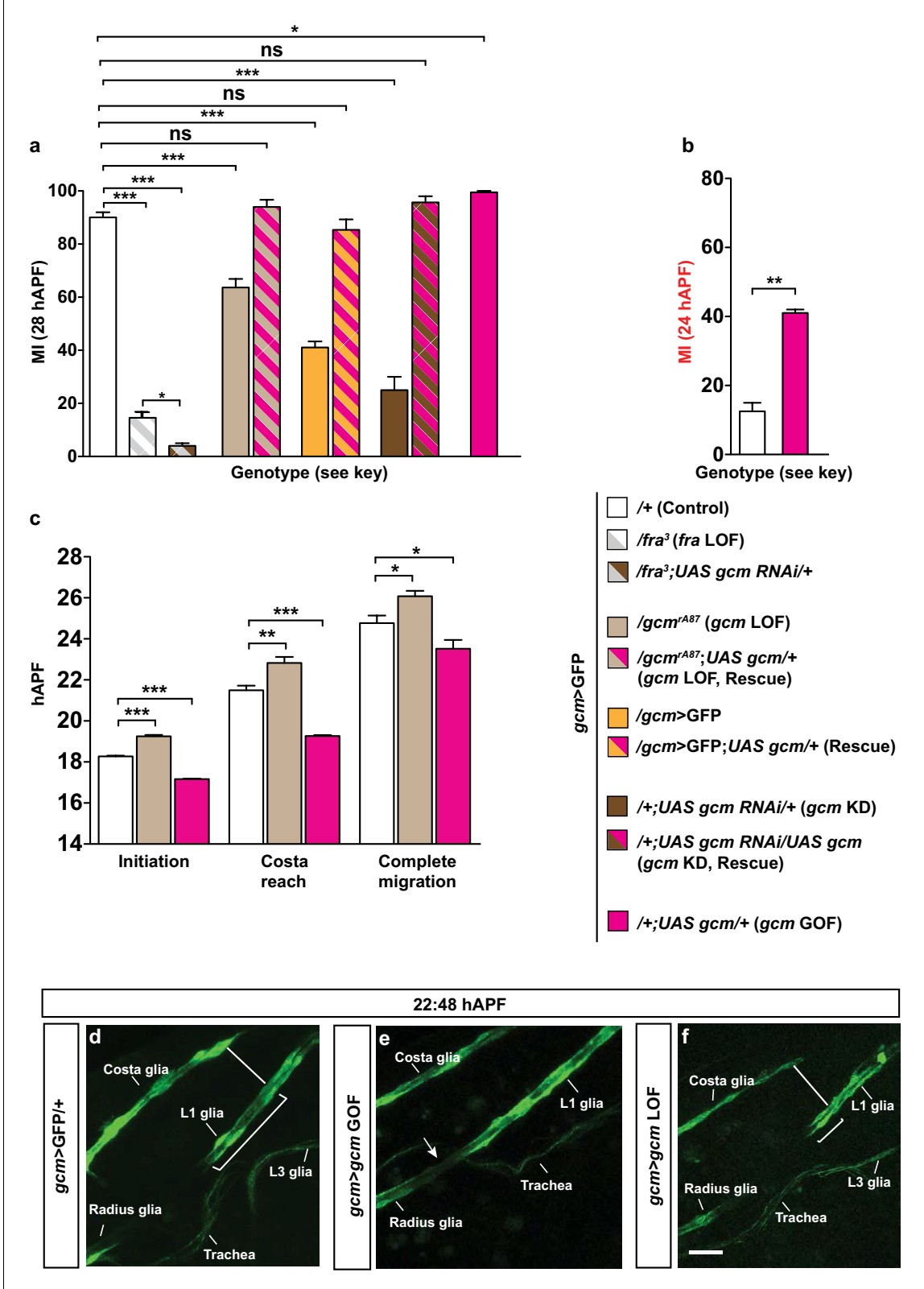

**Figure 4.** Gcm affects collective glia migration in a dose-dependent manner. (**a**) MI in *fra* mutants, *gcm* LOF and rescues of the indicated genotypes: *gcm>GFP/+*, *gcm>gcm* LOF, *gcm>gcm* LOF, *gcm>gcm* KD and *gcm>gcm* GOF wings. (**b**) MI calculated at 24 hAPF in *gcm>gcm* GOF wings. The MI was calculated using the membrane GFP transgenic line. (**c**) Graphical representation of the migratory behavior of *gcm>GFP/+*, *gcm>gcm* LOF and *gcm>gcm* GOF wings at three migratory phases. (**d–f**) Snapshots of a 22:48 hAPF time-lapse analysis on *gcm>GFP/+*, *gcm>gcm* GOF and *gcm>gcm*

*Figure 4 continued on next page*

*Figure 4 continued*

LOF wings. (d) 22:48 hAPF corresponds to the time at which L1 glia have surpassed the level of the Costa (white line) in the control *gcm*>GFP/+ wing, which is why this time point was chosen to compare the position of L1 glia in the different genetic backgrounds. By this time, (e) glia migration is already complete in *gcm*>*gcm* GOF wing (white arrow), while in (f) *gcm*>*gcm* LOF wing, L1 glia are still at the level of Costa (white line). Scale bar: (e–g), 10 µm.

The following source data is available for figure 4:

**Source data 1.** Summary of the role of *gcm* in glia migration.

regulatory role in migration and that the RNAi effects are specific (*Figure 4a*, patterned pink-dark brown column). Furthermore, overexpressing Gcm using the *UAS-gcm* transgene (*gcm* GOF) is sufficient to increase the migration efficiency of glial cells, as the percentage of wings showing complete migration increases compared to that in control animals (*Figure 4a*, pink column and *Figure 4b*).

To determine which migratory step is affected by Gcm, we analyzed the *gcm* LOF (*gcm*>GFP/*gcm*$^{rA87}$) and the *gcm* GOF (*gcm*>GFP/+;*UAS-gcm*/+) wings by confocal time-lapse microscopy and found that migration starts later in Gcm LOF and earlier in *gcm* GOF wings compared to that observed in control wings (*Figure 4c–f*). Thus, Gcm affects the initiation of glial cell migration, like Fra, and it does so in a dose-dependent manner: *gcm* GOF enhances the efficiency of this step and *gcm* LOF lowers it.

In summary, the levels of Gcm and Fra are crucial for the initiation of glia migration.

## Gcm affects migration independently of its role as a fate determinant

The glial to neuron conversion described in *gcm* mutant flies prompted us to ask whether this defect could impact the glial migratory process indirectly (*Hosoya et al., 1995*; *Vincent et al., 1996*; *Jones et al., 1995*; *Bernardoni et al., 1997*). We inspected the rate of glia to neuron conversion in *gcm* KD wings by using the anti-Elav antibody, which specifically recognizes neurons at the analyzed stages (*Figure 5—figure supplement 1a–b'' gcm* KD). Only a minor fraction of the wings contains converted cells and only a few cells are converted into neurons, strongly suggesting that fate conversion is not the cause of the altered MI (on average, 14% of the *gcm* KD wings show up to 10 Repo/Elav double-positive cells along the L1 nerve, ≥15 wings were analyzed per genotype; none were observed in wild-type wings). In addition, we restrained our analysis to *gcm* KD wings that do not show Repo/Elav positive cells and still found a strong delay in migration (MI = 45%).

These data strongly suggest that the glial migration phenotype observed in *gcm* mutant animals is not due to its early requirement in glial cell determination, but specific to cell migration.

## Gcm affects migration independently of glial cell number

Another cause for the migratory phenotype observed in the *gcm* mutant wings might be the control exerted by Gcm on the total number of glial cells. It is indeed plausible that the number of cells in the collective somehow affects the mechanical forces that control migration efficiency, for example through the amount of chemoattractant receptor. The number of Repo-positive cells in *gcm* LOF and KD backgrounds is indeed lower than that in the wild-type glial chain and, accordingly, *gcm* GOF wings contain supernumerary glial cells (*Figure 5a*, light brown, dark brown and pink columns, respectively).

To assess the impact of glial cell number in migration efficiency more directly, we analyzed wings that overexpress proteins promoting or repressing cell division. The exit from the cell cycle results from the timely inactivation of the Cyclin-dependent protein kinase (Cdk) and Cyclin E (Cyc E) complexes. On the one hand, String/Cdc25 encodes a phosphatase that triggers mitosis by activating the Cdc2 kinase, hence enabling cell proliferation (*Edgar et al., 1994*; *Edgar and O'Farrell, 1989*; *Lasko, 2013*). On the other hand, Dacapo functions as an inhibitor of the Cdk–Cyc E complex both in vivo and in vitro, ultimately leading to cell cycle arrest (*de Nooij et al., 1996*; *Lane et al., 1996*). First, we produced animals overexpressing String or Dacapo in glia (*gcm*>GFP) and verified that this induces a significant change in glial cell number compared to that present in control wings (*Figure 5a*, compare white column with columns with vertical and horizontal lines). Then, we

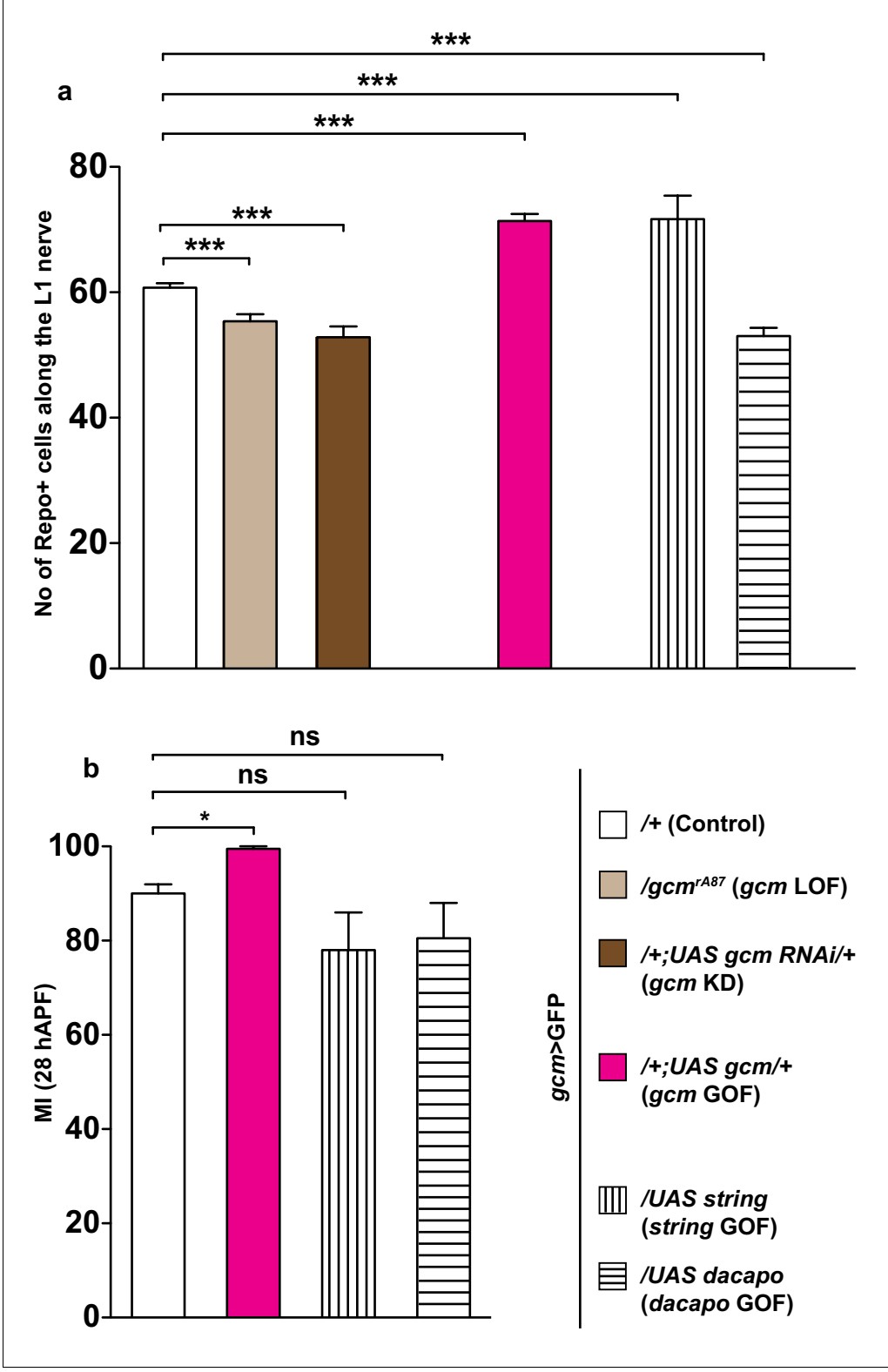

**Figure 5.** The effects of Gcm on collective glia migration are independent of the number of Repo-positive nuclei. (a) Bar chart showing the numbers of L1 glial nuclei in the various genotypes. (b) MI in the various genotypes, calculated using the membrane GFP transgenic line.

The following source data and figure supplement are available for figure 5:

*Figure 5 continued on next page*

*Figure 5 continued*

**Source data 1.** Effects of *gcm* on glia migration are independent of any change in the number of repo+ cells.
**Figure supplement 1.** Fate conversion and Gcm expression in blood cells do not explain the *gcm* migratory phenotype.

analyzed the migration efficiency in both backgrounds and found that it is not affected (*Figure 5b*), even though the glial number increase induced by String overexpression and the decrease induced by Dacapo overexpression are comparable to those observed in *gcm* GOF and LOF, respectively. Finally, we found defective glia migration even in *gcm* KD wings containing a wild-type number of glial cells, in agreement with the above data (*Figure 5—figure supplement 1c*).

Thus, the absolute number of glia does not affect migration efficiency, further corroborating the hypothesis that Gcm specifically affects this process.

## *gcm* expression in blood cells does not affect glial migration

Another round of evidence supports the hypothesis that *gcm* expression in glia, as opposed to effects originating in other cells, is absolutely necessary for their migration . It is already known that *gcm* is also expressed in hemocytes (*Bernardoni et al., 1997*; *Bataillé et al., 2005*; *Waltzer et al., 2010*). So, we checked whether the severe migratory delay is specific to reducing the amount of *gcm* in glia or in hemocytes. To clarify this, we used an independent hemocyte driver, *collagen-Gal4*, which is expressed in the embryonic hemocytes (*Asha et al., 2003*). We knocked down *gcm* in hemocytes (*collagen-Gal4* crossed with the *UAS-gcm-RNAi*) and found no defect in glia migration (*Figure 5—figure supplement 1d*). Furthermore, as Gcm is only present embryonically in blood cells,we used a *gcm-Gal4, tubulin-Gal80thermosensitive(ts)* line to specifically knock down *fra* in glia after embryogenesis and confirmed that, in these conditions too, glia migration is affected (*Figure 5—figure supplement 1e*).

Taken together, these data suggest that Fra and Gcm act cell autonomously in glial cells to control migration on the developing wing disc.

## fra is a direct Gcm target

The gene-expression profile of Fra and the observed genetic interaction between Fra and Gcm prompted us to ask whether Gcm acts on glia migration by inducing Fra expression directly. A DNA adenine methyltranferase identification (DAM ID) screen aimed at identifying the direct targets of Gcm does indeed suggest that this potent transcription factor may directly control the expression of genes involved in glia migration, including *fra* (*Cattenoz et al., 2016*). There are three canonical Gcm-binding sites (GBS) in the *fra* locus, two of which are located at the position of a strong DAM ID peak, which is indicative of Gcm binding (*Figure 6a*). To confirm this data, we assessed whether *fra* expression is activated by Gcm in *Drosophila* S2 cells. To do so, we built a GFP reporter under the control of the *fra* locus containing two GBSs (*Figure 6a–a'*). qRT PCR assays clearly show an increase in the GFP levels upon co-transfection of the reporter vector with the Gcm expression vector and this effect is dose dependent (*Figure 6b*, columns with a red color gradient). To further demonstrate that the effect of Gcm on *fra* is direct (*Figure 6a''*), we showed that Gcm-dependent activation of the reporter is completely abolished upon mutagenesis of the two GBSs (*Figure 6b*, columns with a yellow color gradient). The levels of transfected Gcm were verified by qRT PCR and those of GFP were confirmed by Western blot assays (*Figure 6—figure supplement 1a–c*).

Finally, we complemented the in vitro data with two in vivo assays. First, we showed that Fra levels are affected in opposite direction in *gcm* LOF and GOF wings (*Figure 6c–f*). In *gcm* GOF wings, the levels of the Fra protein already increase by 15 hAPF, in agreement with the observed precocious initiation of migration (*Figure 6—figure supplement 1d–f*, compare with *Figure 3l–o''*). Second, we hypothesized that Fra may constitute an important target of Gcm in L1 glia migration and showed that overexpressing Fra in *gcm* KD wings is sufficient to completely reverse the migratory phenotype that results from *gcm* KD (*Figure 6g*, patterned dark blue and brown columns and *Figure 6h–j*).

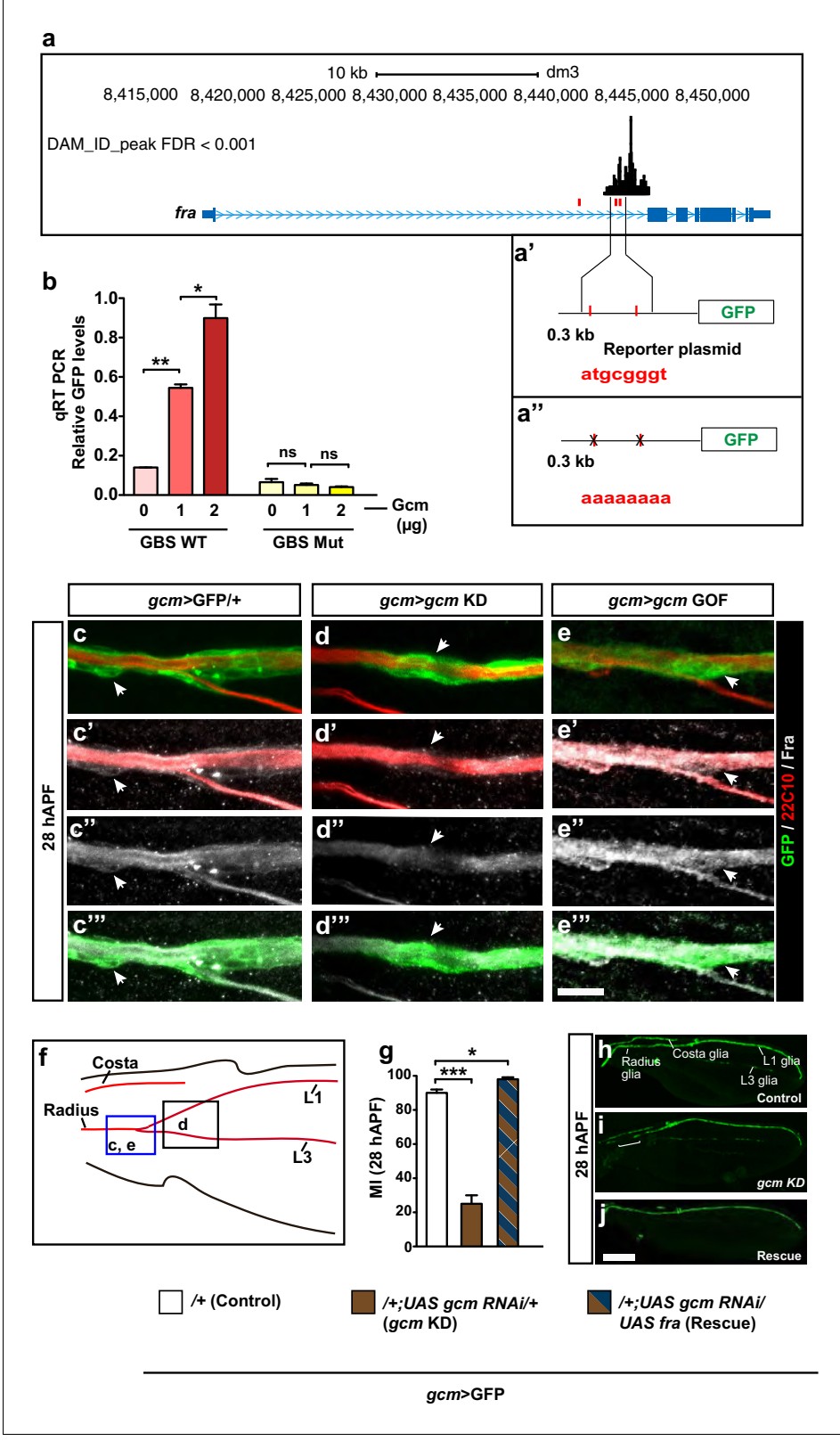

**Figure 6.** *fra* is a direct Gcm target. (a) Schematic representation of the *fra* gene indicated in blue. Lines indicate the introns, thin regions indicate the untranslated sequences, thick regions indicate the coding exons, pale blue arrowheads indicate the direction of transcription. GBSs are indicated in red, the black histogram above shows the *gcm* DAM ID peak at the indicated genomic coordinates. (a') Represents the two GBSs that were amplified and

*Figure 6 continued on next page*

*Figure 6 continued*

put in front of the GFP reporter to generate a *fra* WT plasmid. (**a"**) The two GBSs were mutated in order to generate a *fra* Mut plasmid. (**b**) qRT-PCR analysis of the GFP levels using a WT (red) or a mutant reporter plasmid (yellow) upon co-transfection with increasing doses of the Gcm expression vector (reported as μg) (n=3). (**c–e"'**) Immunolabeling of *gcm>*GFP/+, *gcm>gcm* KD and *gcm>gcm* GOF wings at 28 hAPF using anti-22c10 (red), anti-Fra (gray) and anti-GFP (green). Note the reduced Fra levels in the *gcm>gcm* KD and the increased levels in the *gcm>gcm* GOF wings as compared to those in the *gcm>*GFP/+ wings (arrows). (**f**) Wing schematics: the blue rectangle indicates the region shown in (**c–c"'**) and (**e–e"'**), the black one, the region shown in (**d–d"'**) (**g**) MI of the indicated genotypes calculated using the membrane GFP transgenic line. (**h–j**) 28 hAPF wings from the indicated genotypes: control (*gcm>*GFP/+), *gcm>gcm* KD and rescue (*gcm>gcm* KD/*fra* GOF). (**h**) *gcm>*GFP/+ wing shows complete migration. (**i**) *gcm>gcm* KD wing shows incomplete migration (white bracket). Note that this is a composite image. (**j**) *gcm>gcm* KD/*fra* GOF wing showing complete migration. Scale bar: (**c–e"**), 10 μm; (**h, j**), 80 μm.

The following figure supplement is available for figure 6:

**Figure supplement 1.** Assays confirming *fra* as a direct Gcm target.

In summary, these data suggest that the role of the transcription factor in glial cell migration is through the direct activation of the of chemoattractant receptor, Fra; this in turn implies that an early fate determinant is also capable of directly controlling late developmental events through inducing the expression of cell-surface molecules in a dose dependent manner..

## Role and expression of Netrins

The Fra receptor is known to signal through the two Netrin ligands to mediate cell signaling. To understand which of these ligands were involved in the molecular pathway controlling glial cell migration, we assayed migration in animals that were null mutants for the two netrins: *NetA*$^\Delta$ or *NetB*$^\Delta$ (*Harris et al., 1996*; *Mitchell et al., 1996*; *Brankatschk and Dickson, 2006*). In this experiments, we find that while *NetA*$^\Delta$ mutant glia do not display a defective migratory index (*Figure 7a*, compare gray and blue columns), migration is significantly affected in *NetB*$^\Delta$ mutant glia (*Figure 7a*, compare gray and red columns).

As it has long been known that Netrins can elicit short-range attraction at the *Drosophila* embryonic midline (*Harris et al., 1996*; *Mitchell et al., 1996*; *Brankatschk and Dickson, 2006*), we next checked whether NetB could act in a similar manner in the migrating L1 glia. For this purpose, we used a transgenic line that does not express NetA and that only expresses the wild type or the membrane-tethered form of NetB, which is incapable of signal transduction (*Brankatschk and Dickson, 2006*). The lines *NetA*$^\Delta$ *NetB*$^{TM}$ and *NetA*$^\Delta$ *NetB* had been obtained through homologous recombination and hence express the modified or the wild-type NetB protein at near endogenous levels. Glia migration was found to be comparable to that of wild-type glia in *NetA*$^\Delta$ *NetB*/Y wings, whereas it is delayed in *NetA*$^\Delta$ *NetB*$^{TM}$/Y wings. This suggests that *NetB* must be secreted for proper glia migration to occur (*Figure 7a*, compare light and dark purple columns) implying that the *NetB* ligand functions as a long-range guiding cue for the receptor Fra that is expressed in glial cells.

We next sought to determine the source of the ligand NetB. To do this, we used a transgenic line that is routinely employed as a reporter of NetB expression (*Timofeev et al., 2012*; *Hayashi et al., 2002*). *NP4151>UAS GFP* reports *NetB* expression in the proximal region of the wing (19 hAPF) (*Figure 7b*). This profile of expression fits well with the distal to proximal migration of L1 glia, and we reasoned that if NetB were to act as a chemoattractant, its loss should cause glial migratory defects similar to those induced by the loss of Fra. We therefore knocked down *NetB* by crossing *NP4151-Gal4* to *UAS-NetB-RNAi* (*NetB* KD) flies and uncovered a severe migration defect, which could be rescued by simultaneously expressing *UAS-NetB* and *UAS-NetB-RNAi* (*Figure 7c*, first three columns). Moreover, overexpressing *NetB* in its territory of expression enhances the efficiency of glial cell migration, as revealed by the MI at 24 hAPF (*Figure 7d*). By contrast, ectopic expression of *NetB* in the posterior wing compartment using *engrailed-Gal4* driver (*en>*) (*Hidalgo, 1994*; *Lawrence and Morata, 1976*) or in the distal part of the wing using *GMR 29F05-Gal4* (*GMR*

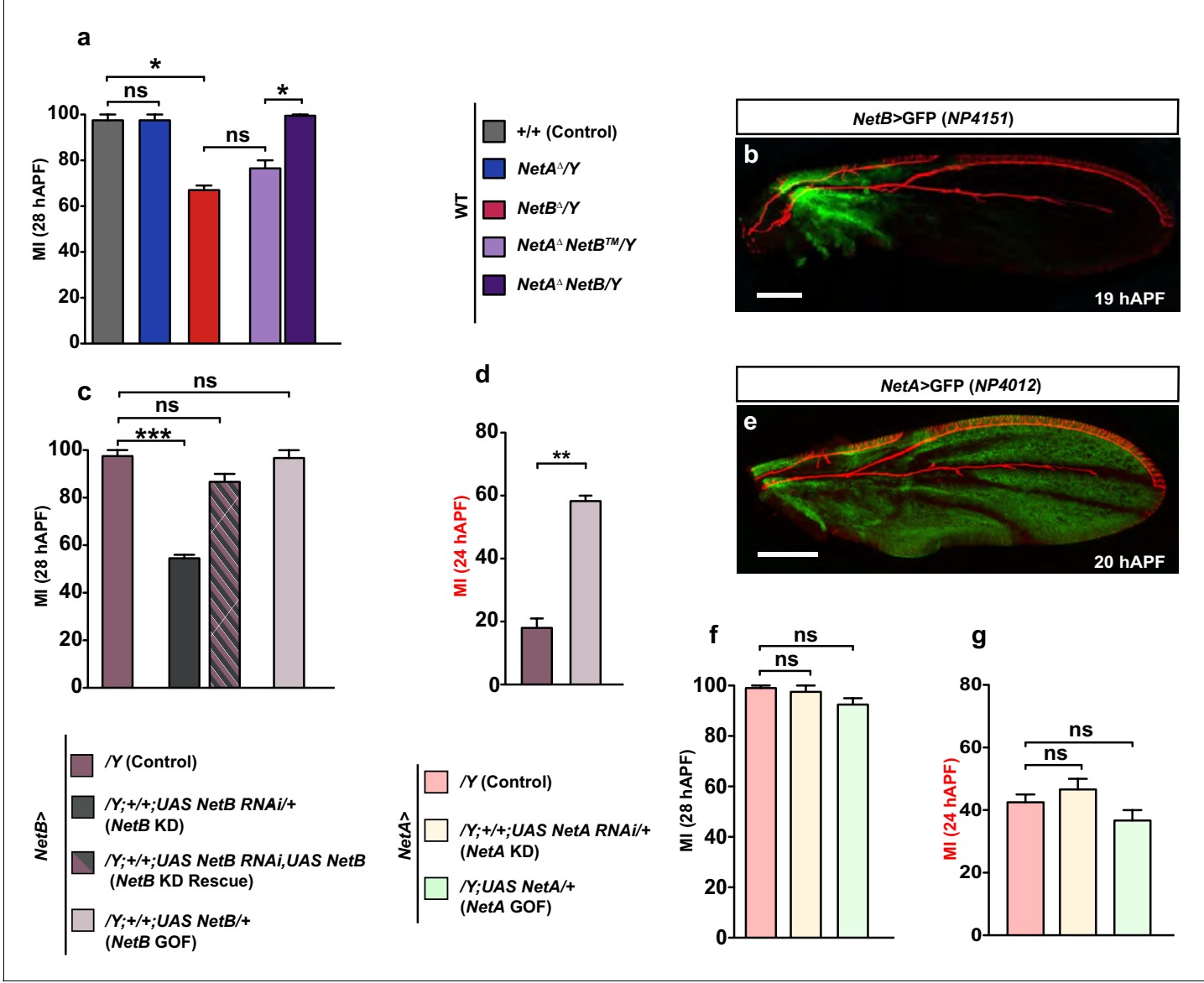

**Figure 7.** NetB serves as a chemoattractant in collective glia migration. (a) MI of the indicated genotypes. Histogram shows the MIs quantified for WT, for $NetA^\Delta$ or $NetB^\Delta$ single mutant wings and for $NetA^\Delta NetB^{TM}$ or $NetA^\Delta NetB^{myc}$ wings. (b) NP4151-Gal4 driven GFP expression of NetB in a 19 hAPF wing. Proximal NetB expression as revealed by the GFP labeling (green). Anti-22C10 is in red. (c, d) Histograms representing the MI of the indicated genotypes. (e) NP4012-Gal4 driven GFP expression of NetA in a 20 hAPF wing. NetA is expressed in the wing epithelium as revealed by the profile of GFP (green). Anti-22C10 is in red. (f, g) Histograms representing the MI of the indicated genotypes. For this figure the MI was calculated by nuclear labeling. Scale bars: (b, e), 80 μm.

The following source data and figure supplements are available for figure 7:

**Source data 1.** Summary of the role of Netrins in glia migration.

**Figure supplement 1.** Role of Netrins in collective glia migration.

**Figure supplement 2.** Early expression of NetB.

*29F05>*)(*Pfeiffer et al., 2008*) does not affect the migration efficiency of glial cells (*Figure 7—figure supplement 1a*).

Finally, NetA is almost ubiquitously expressed in the epithelium, as revealed by the use of the *Gal4* transgenic line *NP4012* (*Figure 7e*) (*Timofeev et al., 2012*; *Hayashi et al., 2002*) and knocking down or overexpressing NetA with that driver has no impact on glia migration at early or at late stages (*Figure 7f,g*). Moreover, NetA overexpression in the NetB expression territory fails to enhance migration efficiency (*Figure 7—figure supplement 1b*) or to rescue the *NetB* KD phenotype confirming our finding that NetA is not involved in the process of glial cell migration (*Figure 7—figure supplement 1c*).

Altogether, our data strongly support the hypothesis that secreted NetB in the proximal wing provides a crucial chemoattractant cue for Gcm-mediated Fra expression, hence controlling the efficiency of glia migration. Our data also suggest that NetB may contribute but is not sufficient to trigger directionality.

## Unc5 controls the late phase of L1 glia migration

*unc5* is a repellant receptor for Netrins and has been previously shown to be transiently expressed and required in the embryonic exit and peripheral glia (PG) associated with both the segmental and intersegmental nerves (*von Hilchen et al., 2010*; *Keleman and Dickson, 2001*; *Freeman et al., 2003*). We therefore asked whether *unc5* might also be involved in the migration of glia in the developing wing. To do this, we first analyzed its spatio-temporal expression profile during development. Unc5 is undetectable in the wing disc at 15 hAPF (*Figure 8—figure supplement 1a–a''*). Its expression is first seen at 18 hAPF at low levels and this expression progressively decreases to completely fade away by 29 hAPF (*Figure 8a–d'''*).

If *unc5* were to act as a repulsive receptor, the efficiency of glia migration would increase if its expression decreases, but neither RNAi-mediated KD of *unc5*, nor the null $unc5^8$ mutation (*Labrador et al., 2005*) affect glia migration efficiency at the early and late stages of development (*Figure 8e*, first three columns; *Figure 8—figure supplement 1b*). Thus, the loss of Unc5 does not seem to enhance the migration efficiency of L1 glia in the developing wing. We then asked whether Unc5 expression must be tightly regulated, and found that overexpressing Unc5 affects the efficiency of glia migration by delaying it, a phenotype that is opposite to the Fra overexpression phenotype (*Figure 8e*, compare blank and green columns). This delayed migration phenotype was rescued by reducing the levels of unc5 using the *unc5* KD construct or the mutant allele $unc5^8$, in this overexpression background. This rescue demonstrates a direct effect of *unc5* on glia migration (*Figure 8e*, patterned green and purple column, *Figure 8—figure supplement 1c*, third column).

Thus, *fra* and *unc5* serve opposite roles in glia migration, with *fra* being necessary to trigger migration and *unc5* delaying the migratory process. The two molecules seem to work in the same signaling pathway because the *unc5* GOF phenotype is further enhanced by lowering the levels of Fra (*Figure 8e*, patterned green and light blue column). Also, the migratory phenotype induced by *unc5* overexpression is rescued by simultaneously overexpressing Fra (the rescue was analyzed at an early stage for a better quantification; *Figure 8f*, patterned green and dark blue column). Finally, knocking down *unc5* rescues the *fra* KD phenotype (*Figure 8—figure supplement 1c*, last two columns).

Altogether, the data described above strongly suggest that Unc5 can act as a repellant but that its expression is not sufficient to affect migration efficiency, which is mostly controlled by Net–Fra interaction. A likely explanation for these results is that Unc5 contributes at late migratory stages, and time-lapse movies indeed show that *unc5* overexpression delays migration but does not affect the initiation step (*Figure 9—figure supplement 1a*). To further test this hypothesis, we assessed the effects of knocking down Unc5 using *repo-Gal4*, which is a late glial driver compared to *gcm-Gal4*. In these conditions, we did see an acceleration in migration completion (*Figure 8g*). Finally, since Unc5 has been identified in microarrays obtained upon overexpressing Gcm (*Freeman et al. 2003*) and in the DAM ID screen performed to identify the direct targets of Gcm (*Cattenoz et al., 2016*), we asked whether Unc5 expression is directly induced by Gcm. Co-transfection assays in S2 cells similar to those performed on *fra* and using an *Unc5–GFP* reporter containing two GBS close to the *unc5* transcription start site, suggest that Unc5 is a rather weak target of

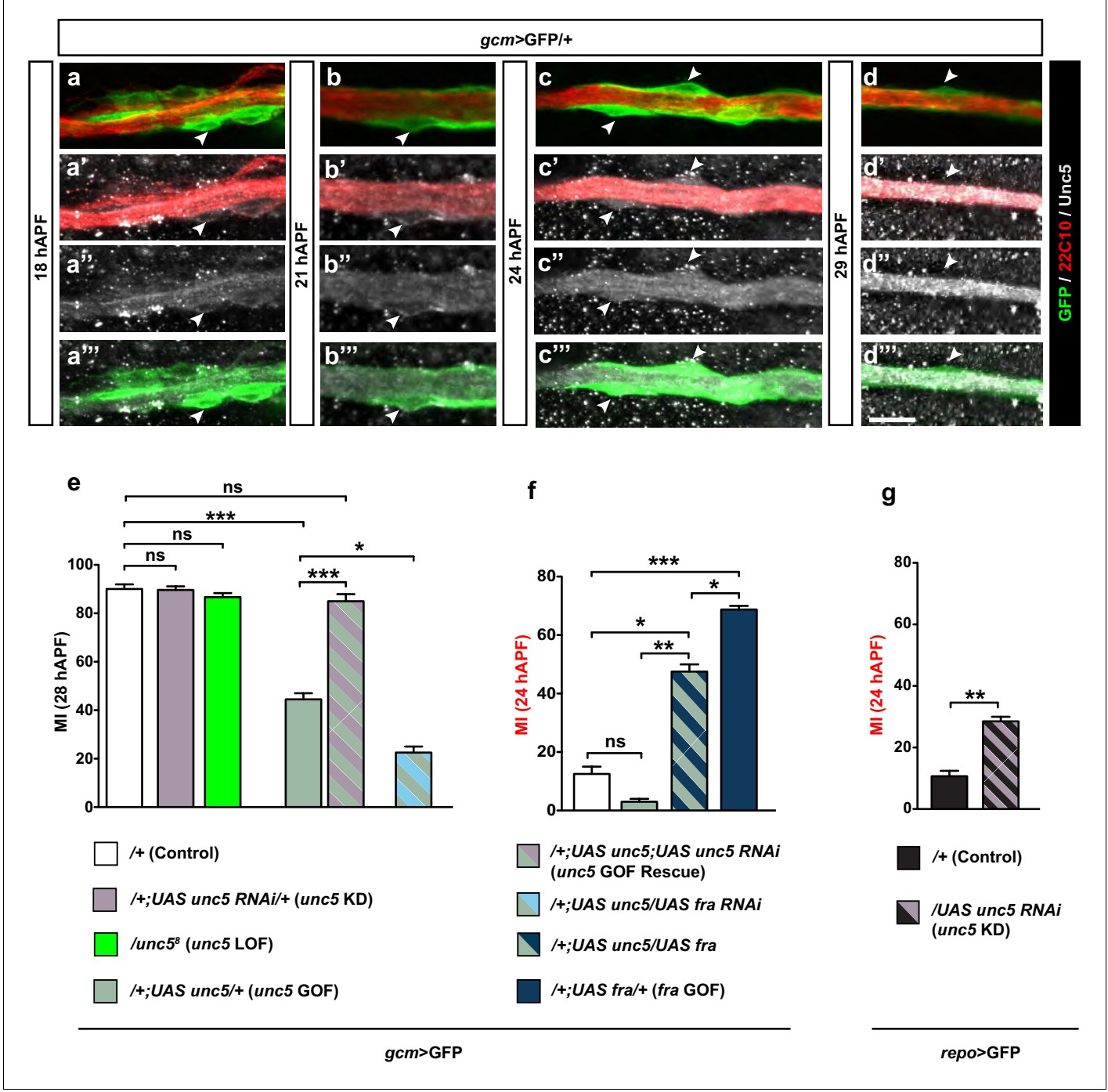

**Figure 8.** Unc5 may act as a repellant in glia migration. (**a–d'''**) Wing immunolabeled with anti-22c10 (red), anti-Unc5 (gray) and anti-GFP (green) in the transgenic line *gcm*>GFP/+ at different migratory stages. Unc5 starts being expressed in glia and neurons at around 18 hAPF and fades away by 29 hAPF. Arrowheads show glia that express Unc5 at weak levels (**a–c'''**) or that do not express Unc5 (**d–d'''**). Note that (**a–d"**) are comprised of a few sections rather than maximum confocal projections. (**e–g**) Histograms representing the MI of the indicated genotypes, which in this figure was calculated using the membrane GFP transgenic line. Scale bars: (**a–d"**), 10 µm.

The following source data and figure supplement are available for figure 8:

**Source data 1.** Summary of role of *unc$^5$* in glia migration.

**Figure supplement 1.** Unc5 in collective glia migration.

Gcm (*Figure 8—figure supplement 1d*). This was further confirmed in *gcm* gain-of-function in vivo assays (data not shown) similar to those mentioned above on *fra*.

In summary, these data suggest that apart from a role for chemoattraction mediated through Fra and NetB, which determines the initiation of glial cell migration, a later chemorepulsion mediated through Fra-Unc5 ensures the termination of migration. Furthermore, the expression of Unc5 is also directly regulated by the transcription factor gcm suggesting that the same early determinant controls the timely expression of effector genes that initiate and terminate glial cell migration.

## Discussion

Collective migration is a complex biological process that allows cells to leave their place of birth and reach their destination in a coordinated and timely manner. Here we dissect the signaling pathway underlying the collective migration of glia on the developing *Drosophila* wing. The careful coordination between extrinsic cues and intrinsic, cell autonomous mechanisms ensure that timely migration of the glia occurs on the wing disc. The chemoattractant receptor Fra controls glia migration in response to a long-distance signal, thechemoattractant NetB. Fra is expressed before glia start to move and triggers migration initiation in a dose-dependent manner. Such tight control of Fra expression depends upon the Gcm transcription factor, which directly induces the expression of Fra at threshold levels. Thus, the glial determinant factor also affects a specific step of collective migration by regulating the key effector gene.

Collective migration comes in different flavors: streams, chains, sheets and clusters, which all imply tight coordination and cell–cell interactions (*Klämbt, 2009*; *Gilmour et al., 2002*; *Gupta and Giangrande, 2014*; *Marín et al., 2010*; *Rørth, 2003*). The small cluster of *Drosophila* border cells migrates through nurse cells towards the oocyte in response to growth factors (*Montell, 2003*; *Rørth, 2009*). The stream of hundreds of proliferating cells of the fish lateral line migrate directionally upon expressing different chemokine receptors within the collective (*Dambly-Chaudiere et al., 2007*; *David et al., 2002*; *Ghysen and Dambly-Chaudière, 2004*; *Haas and Gilmour, 2006*). The current challenge is to dissect the role of the signaling pathways in the different steps (initiation, maintenance and arrest) and features of collective migration (adhesion, overall velocity and timing) and to analyze this in vivo. The second challenge is to understand how those signaling pathways are regulated, an issue that has been addressed at the post-transcriptional level (*Yu and Bargmann, 2001*) but much less so at the transcriptional level.

By focusing on the Netrin signaling pathway, which has been extensively studied in the context of axonal navigation and cell proliferation, we have addressed the above questions using the migrating wing glia of *Drosophila.* The analysis of the mutant phenotypes and the time-lapse approach show that the Fra receptor controls glia migration along the L1 nerve. Reducing the amount of Fra delays migration, whereas excessive Fra in the glial cells triggers their precocious migration. Thus, the Fra receptor plays an instructive role in the first step of collective migration, initiation. These data highlight the importance of quantitative regulation: large cohorts of cells are likely to need strong forces to switch from an immotile to a motile phenotype; therefore only the strong expression of the receptor allows migration toward the chemoattractant. Other migratory collectives also depend on quantitative regulation. Typically, epithelial cells migrating in groups exert much stronger forces than an individual cell before and after epithelial-mesenchymal transition (*du Roure et al., 2005*). The cell adhesion molecule E-cadherin plays an important role in supporting the directional and efficient collective migration of epithelial cells by mediating adhesion and force generation (*Li et al., 2012*). Additionally, the collectively migrating border cell cluster cannot exert enough adhesive/pulling force to move between the nurse cells in the absence of E-cadherin (*Niewiadomska et al., 1999*).

The timely regulation of the Fra receptor and its early role in collective migration allowed us to show that the Gcm fate determinant affects migration through this receptor. Thus, early genes not only trigger the expression of transcription factors that in turn implement a specific developmental program, but also directly contribute to the acquisition of specific phenotypes, such as the migratory potential. Similarly, the Lim-homeodomain transcription factor Islet was shown to specify the electrical properties of motor neurons by repressing the expression of the ion channel Shaker during development, suggesting that the regulation of late genes by early transcription factors might also be a common phenomenon (*Wolfram et al., 2012*). These observations prompt us to revisit the role of the so-called 'master regulators', which indeed control processes beyond fate choices.

Netrins are thought to act either in a gradient at long range, as secreted molecules, or at short range, as membrane-tethered molecules (*Lai Wing Sun et al., 2011*; *Brankatschk and Dickson, 2006*). In contrast with studies in the *Drosophila* embryo and visual system (*von Hilchen et al., 2010*; *Timofeev et al., 2012*; *Brankatschk and Dickson, 2006*), we observed that secreted NetB in the developing wing acts at a long range: glia migration is delayed when solely membrane-tethered NetB is available at near-endogenous levels. While we still do not know whether the distinction between long- and short-range signaling depends on the complement of surface receptors and the associated transduction pathways, it seems that different strategies are used in specific processes. Future studies will reveal whether long-range signaling is specifically dedicated to migration over many cell diameters and/or to large collectives.

NetA and NetB have been suggested to act on the embryonic longitudinal glia through Fra (*von Hilchen et al., 2010*) and both Netrins are reported to act as chemoattractants, even though in one case, only NetB was proposed to mediate dendritic targeting via Fra (*Matthews and Grueber, 2011*). Clearly, downregulating or overexpressing NetA has no impact on the migrating glial cells of the wing and NetA cannot rescue the NetB phenotype. While it is possible that NetA may affect other aspects of wing glia biology, our data indicate that NetB alone serves as a chemoattractant ligand to guide L1 glia migration upon signaling to Fra. Studies in other systems will assess whether it is only NetB that works as a long-range chemoattractant. If this were the case, it would be interesting to assess whether the different behavior of NetA and NetB relies on the intrinsic potentials of the two ligands (e.g. the affinity with which NetA and NetB bind to their receptors) or on extrinsic cues (e.g. cell-specific cofactors modulating NetA and/or NetB activity).

The chemoattractant receptor DCC/Fra and the chemorepellant receptor Unc5 have been mostly studied in different cell types, so it is unclear whether the same process and cells require the counteracting activities of these molecules. While mammalian migrating oligodendrocyte precursor cells express both DCC and Unc5, the precise function of these receptors has not been assessed (*Tsai et al., 2003*). We found that the two receptors Fra and Unc5 are expressed by the same set of PNS glial cells, with Fra appearing earlier than Unc5. The two receptors seem to have a different impact on wing glia migration and their stoichiometry is an important factor, as reducing Unc5 levels does have an effect on migration when *fra* is downregulated (*Figure 8—figure supplement 1b* last columns). The fact that overexpressing Unc5 and Fra have opposite effects suggests that Unc5 converts the Netrin-mediated signal from attraction to repulsion in the *Drosophila* wing glia, in agreement with published data on chimeric molecules. For example, when fused to the cytoplasmic domain of Unc5, the extracellular domain of DCC is as effective in mediating the chemorepellant response to Netrin as a wild type Unc5 (*Hong et al., 1999*). Additionally, a study performed on *Drosophila* showed that fusing the cytoplasmic domain of Fra with the extracellular domain of Unc5 signals chemoattraction (*Keleman and Dickson, 2001*). This may explain how the same cues such as Netrins can work differently on various aspects of collective migratory behavior.

The key element in this study is time: Gcm accumulates and triggers the onset of glial migration through *fra* but subsequently fades away. The fact that the expression of *fra* stays on till the end of migration suggests that another player, possibly a direct target of Gcm, maintains its expression. Being a major Gcm target, *repo* represents a potential candidate. Our in vivo and cell transfection data using a Repo expression vector are in line with this hypothesis and suggest that Repo may play a regulatory role in the maintenance of *fra* at late migratory stages (*Figure 9a*). It is known that Repo expression first depends on Gcm but subsequently becomes independent of it through autoregulation (*Flici et al., 2014*), indicating a transition from early to late events in gliogenesis (see model in *Figure 9b*). We propose that the initial Fra expression in wing glia strictly depends on Gcm and triggers the initiation of migration (see the graph in *Figure 3—figure supplement 1e*). Accordingly, the overexpression of Fra using the *repo* promoter does not induce migratory defects (*Figure 9c,d*) and *repo>fra* KD wings show only modest defects in migration initiation (*Figure 9—figure supplement 1b*). Later on, however, when Gcm fades away, Repo takes over in controlling Fra expression, hence affecting later phases of migration (*Figure 9—figure supplement 1b*, see model in *Figure 9b*). Conversely, the *unc5* repellant is a weak target of Gcm thatmay require Repo and controls migration completion.

With respect to what has previously been described in embryonic glia migration, we propose a different role for Netrins and their receptors in the migration of L1 glia. The migration of embryonic longitudinal glial cells seems to rely on very early expression of Fra, which is detected in the

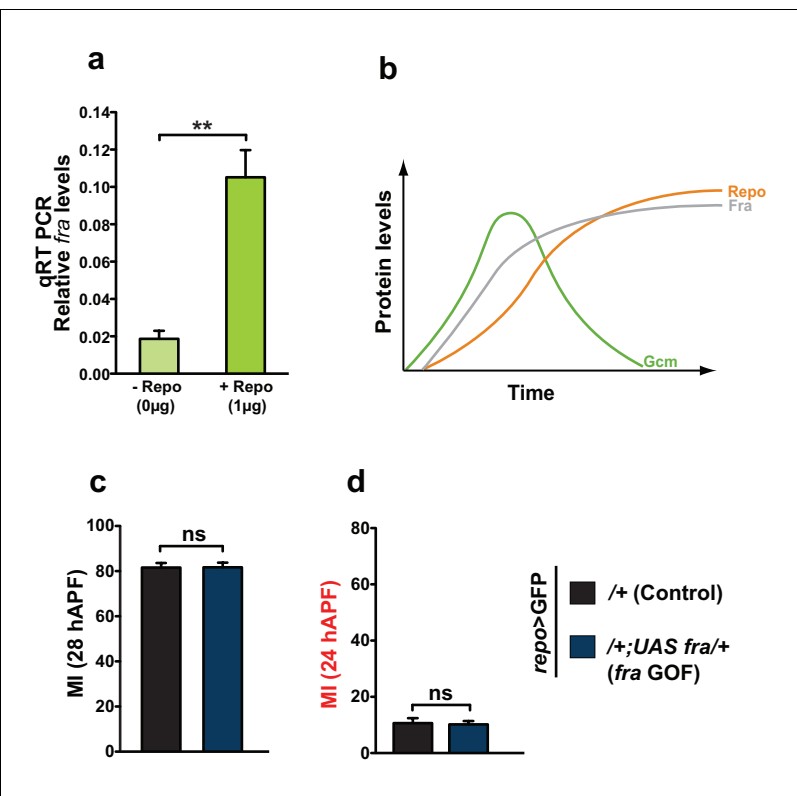

**Figure 9.** Repo regulates Fra at late stages. (**a**) Histogram showing the endogenous expression of *fra* upon S2 cell transfection with a Repo expression vector. The y-axis represents the relative expression levels in cells transfected with Repo compared to that observed in cells not transfected with the Repo expression vector (n=3). (**b**) Schematic summarizing the regulatory network. (**c, d**) Graphs representing the MI found at early and late stages in *fra* GOF animals crossed with the *repo-Gal4* line. In this figure, the MI was calculated using the membrane GFP transgenic line.

The following source data and figure supplement are available for figure 9:

**Source data 1.** Summary of role of *repo* in glia migration.

**Figure supplement 1.** In vivo analysis of Unc5 GOF wings and *fra* KD wings using a late driver.

---

longitudinal glial precursor, but no longer in the longitudinal glial cells (*von Hilchen et al., 2010*; *Kolodziej et al., 1996*). In our work, we do detect Fra expression in the wing glia. However, while in the case of the longitudinal glia both Netrins guide the glia toward the midline, the L1 glia seem to require only NetB. Moreover, ectopic expression of Netrin affects the migration of the longitudinal glia, whereas it does not do so in wing glia, hence suggesting that Netrin has a role in migration efficiency rather than on directionality. And finally, we propose that Fra and Unc5 act in different migratory phases of the same glial population, the first phase being under the direct control of the Gcm glial determinant. Overall, this paper displays a model in which the ligand NetB is expressed early during wing development (*Figure 7—figure supplement 2*), whereas its receptor Fra is expressed right before migration onset and cell autonomously decides the time of migration initiation of the glial chain. Unc5 appears later and contributes to migration termination. The acquisition of the Fra levels that trigger migration is part of the cell-specification program dictated by the Gcm glial determinant, which not only induces the expression of downstream transcription factors but also directly implements a specific developmental program by triggering the expression of effector genes.

Together, these findings in the *Drosophila* wing suggest that the dynamic and coordinated action of fate determinants and chemotropic cues contribute to the timely and efficient migration of the

glial cells. A similar molecular mechanism, relying on Netrins or other localized attractive cues and their receptors, may be used in other cases of collective migration.

## Materials and methods

### Fly stocks and genetics

Fly stocks were raised at 25°C in standard medium. *repo-Gal4* (indicated as *repo>)* was used to detect glial-specific expression of *UAS*-PH-GFP (fusion protein between the Pleckstrin homology domain of PLC-d and the GFP coding sequence). *gcm-Gal4 UAS*-mCD8-GFP (membrane localization) (*gcm>GFP*) was used as an early glial-specific driver (**Jacques et al., 2009**). *gcm-Gal4,tub-Gal80$^{ts}$* (**Soustelle et al., 2007**). The other strains used were *gcm 6Kb> 42*; *fra$^3$* (**von Hilchen et al., 2010**); *UAS-fra* (**von Hilchen et al., 2010**); *UAS-fra-RNAi* (**Manhire-Heath et al., 2013**); *UAS-fra$^{\Delta}$C* (**Garbe et al., 2007**); *gcm$^{rA87}$* (**Vincent et al., 1996**); *gcm$^{N7-4}$* (**Vincent et al., 1996**); *UAS-gcm (F18A)* (**Bernardoni et al., 1998**); *UAS-gcm-RNAi*; *gcm>GFP/gcm>GFP* (used as a homozygous mutant of *gcm*) (**Popkova et al., 2012**); *UAS-string* (**Inaba et al., 2011**); UAS-dacapo (**Lane et al., 1996**); *gcm$^{34}$* (**Vincent et al., 1996**); *NetA$^{\Delta}$* (**von Hilchen et al., 2010**; **Newquist et al., 2013a, 2013b**); *NetB$^{\Delta}$* (**von Hilchen et al., 2010**; **Newquist et al., 2013a, 2013b**); *NetA$^{\Delta}$ NetB$^{TM}$* (**Brankatschk and Dickson, 2006**); *NetA$^{\Delta}$ NetB$^{myc}$***Brankatschk and Dickson, 2006**) (note that both *NetA$^{\Delta}$ NetB$^{TM}$* and *NetA$^{\Delta}$ NetB$^{myc}$* encode the c-myc epitope tags); *UAS-NetB-RNAi* (**Manhire-Heath et al., 2013**); *UAS-NetB* (**Timofeev et al., 2012**); *NP4151-Gal4* and *NP4012-Gal4* (DGRC, Kyoto) (**Timofeev et al., 2012**; **Hayashi et al., 2002**); *UAS-NetA-RNAi* (**Manhire-Heath et al., 2013**); *UAS-NetA* (**Newquist et al., 2013a, 2013b**); *unc5$^8$* (**Labrador et al., 2005**); *collagen-Gal4* (**Asha et al., 2003**); *engrailed-Gal4* driver (**Hidalgo, 1994**; **Lawrence and Morata, 1976**); *GMR 29F05-Gal4* (**Pfeiffer et al., 2008**); *UAS-unc5-RNAi*; *UAS-unc5* (**von Hilchen et al., 2010**); *nsyb-Gal4>GFP-LAMP* (B# 42714); *elav-Gal4*. The RNAi lines were obtained from Bloomington and/or the VDRC stock center.

To generate the coinFLP clones, the *hsFLP;UAS-fra-RNAi* built with *hsFLP* (B# 6) and *UAS-fra-RNAi* (**Manhire-Heath et al., 2013**), was crossed with *UAS-mCD8-RFP,LexAop-mCD8-GFP;CoinFLP-LexA.GAL4* (B# 58754) (**Bosch et al., 2015**). Heat shock was carried out in wandering third instar larvae at 37°C for 30 min, wings from female pupae were dissected at 18 hAPF.

The summary table here below provides the genotypes for each figure reporting MI, for the sake of simplicity.

| Genotypes | Abbreviated as |
| --- | --- |
| *Figure 2* | |
| WT | Control |
| *fra3/+* | LOF |
| *repo>*GFP/+ | Control |
| *repo>*GFP/*fra3* | *fra* LOF |
| *Figure 3* | |
| *gcm>*GFP/+ | Control |
| *gcm>*GFP/+;*UAS fra RNAi/+* | *fra* KD |
| *gcm>*GFP/+;*UAS fra RNAi/UAS fra* | *fra* KD, Rescue |
| *gcm>*GFP/+;*UAS fra/+* | *fra* GOF |
| *gcm>*GFP/+*UAS fra$^{\Delta}$C* | |
| *Figure 4* | |
| *gcm>*GFP/+ | Control |
| *gcm>*GFP/*fra3* | *fra* LOF |
| *gcm>*GFP/*fra3;UAS gcm RNAi/+* | |
| *gcm>*GFP/*gcmrA87* | *gcm* LOF |
| *gcm>*GFP/*gcmrA87;UAS gcm/+* | *gcm* LOF, Rescue |

| | |
|---|---|
| *gcm*>GFP/*gcm*>GFP | |
| *gcm*>GFP/*gcm*>GFP;*UAS gcm*/+ | Rescue |
| *gcm*>GFP/+;*UAS gcm RNAi*/+ | *gcm* KD |
| gcm>GFP/+;*UAS gcm RNAi/UAS gcm* | *gcm* KD, Rescue |
| gcm>GFP/+;*UAS gcm*/+ | *gcm* GOF |
| ***Figure 5*** | |
| *gcm*>GFP/+ | Control |
| gcm>GFP/gcmrA87 | *gcm* LOF |
| *gcm*>GFP/+;*UAS gcm RNAi*/+ | *gcm* KD |
| gcm>GFP/+;*UAS gcm*/+ | *gcm* GOF |
| *gcm*>GFP/*UAS string* | *string* GOF |
| *gcm*>GFP/*UAS dacapo* | *dacapo* GOF |
| ***Figure 6*** | |
| *gcm*>GFP/+ | Control |
| *gcm*>GFP/+;*UAS gcm RNAi*/+ | *gcm* KD |
| *gcm*>GFP/+;*UAS gcm RNAi/UAS fra* | Rescue |
| ***Figure 7*** | |
| WT | Control |
| NetAΔ/Y | |
| NetBΔ/Y | |
| NetAΔNetBTM/Y | |
| NetAΔNetB/Y | |
| NetB>/Y | Control |
| NetB>/Y;+/+;UAS NetB RNAi/+ | *NetB* KD |
| NetB>/Y;+/+;UAS NetB RNAi/UAS NetB | *NetB* KD, Rescue |
| NetB>/Y;+/+;UAS NetB/+ | *NetB* GOF |
| NetA>/Y | Control |
| NetA>/Y;+/+;UAS NetA RNAi/+ | *NetA* KD |
| NetA>/Y;UAS NetA/+ | *NetA* GOF |
| ***Figure 8*** | |
| *gcm*>GFP/+ | Control |
| *gcm*>GFP/+;*UAS unc5 RNAi*/+ | *unc5* KD |
| gcm>GFP/unc5[8] | *unc5* LOF |
| *gcm*>GFP/+;*UAS unc5*/+ | *unc5* GOF |
| *gcm*>GFP/+;*UAS unc5; UAS unc5 RNAi* | *unc5* GOF, Rescue |
| *gcm*>GFP/+;*UAS unc5/UAS fra RNAi* | |
| *gcm*>GFP/+;*UAS unc5/UAS fra* | |
| *gcm*>GFP/+;*UAS fra*/+ | *fra* GOF |
| *repo*>GFP/+ | Control |
| *repo*>GFP/*UAS unc5 RNAi* | *unc5* KD |
| ***Figure 9*** | |
| *repo*>GFP/+ | Control |
| *repo*>GFP/+;*UAS fra*/+ | *fra* GOF |

## Molecular cloning

For the *fra* gene, oligonucleotides (see primers sequence below) surrounding the GBSs were designed with flanking restriction sites for KpnI at the 5' extremity and NHeI at the 3' extremity. Each pair of oligonucleotides was used to amplify the genomic region encompassing the GBSs using the Expand High fidelity polymerase (Roche). The amplicons were digested with 20 U of KpnI (NEB # R3142S) and 20 U of NheI (NEB # R3131S) in Cutsmart buffer (NEB # B7204S) for 2 hr min at 37°C. The digested amplicons were then cleaned using the PCR clean-up kit (Macherey-Nagel (MN) # 740609) according to manufacturer's instructions.

For ligation, 50 ng of the digested probe were used and cloned into the pGreen Pelican vector overnight at 18°C. 1 µL of the ligated product was used for transformation of electro competent DH5α bacterial cells. Bacteria were then kept for 1 hr at 37°C and plated on ampicillin-containing medium. After overnight incubation at 37°C, several colonies were picked up in separate tubes containing LB and incubated overnight at 37°C. The following day, plasmid DNA was extracted using the DNA Purification Kit (MN #740410) according to the manufacturer's instructions; DNA from positive colonies was identified upon gel electrophoresis and sent for sequencing for final confirmation.

Same procedure was conducted to build the mutated *fra* reporter plasmid.

Following oligonucleotides were used:

*fra* WT forward:

5' GAGAGGTACCGTGTCCAAAAATGCGGGTCTGTTTCTCG 3'

*fra* WT reverse:

5'GAGAGCTAGCGTTAAGACAAACATGCAGGCATAAAGACATG 3'

*fra* Mutant forward:

5'GAGAGGTACCGTGTCCAAAAAAAAAACTGTTTCTCGAAATTGAGTT 3'

*fra* Mutant reverse:

5'GAGAGCTAGCGTTAAGACAAACAAAAAAAAATAAAGACATGAAATGGATG 3'

Similarly, we built *unc5* WT and mutant plasmids with flanking restriction sites for EcoR1 at the 5' extremity and Kpn1 at the 3' extremity.

Following oligonucleotides were used:

*unc5* WT forward:

5' GAGAGAATTCTCGTTTTCCCGTTTAGGGCA 3'

*unc5* WT reverse:

5' GAGAGGTACCACTAGCGCTCACCACAGTTC 3'

*unc5* Mutant forward 1:

5' GTGTGAACAGTGATATAAAGTGCACCGTGTAAAAAAAATAGAGATACCT 3'

*unc5* Mutant reverse 1:

5' TTCGTGTGGCACTAGGTTAGGTATCTCTATTTTTTTTTACACGGTGCACT 3'

*unc5* Mutant forward 2:

5' ATAAAAACAAGCCGCACACACAGTAGCACAAAAAAAAAAAAGGGGCGCAC 3'

*unc5* Mutant reverse 2:

5' CATCGGACGACCACTGCAGTGCGCCCCTTTTTTTTTTTTGTGCTACTGTG 3'

## Co-transfection and Western blot assays

Co-transfections in S2 cells were carried out using Lipofectamine (Invitrogen). $6 \times 10^6$ cells were cultured in 6-well plates containing Schneider medium. In each well, cells were transfected with 1 µg of *fra* WT or

mutant reporter plasmid, 1 µg of *pPAC-lacZ* (*Flici et al., 2014*; *Cattenoz et al., 2016*) as a transfection control, 0.5 µg or 1 µg or 2 µg of *pPac gcm* (*Cattenoz et al., 2016*) expression vector and *pPac* 'empty' (*Flici et al., 2014*) to make up the volume up to 4 µg. Cells were collected 48 hr after transfection, washed in cold PBS and resuspended in lysis buffer (25 mM Tris-phosphate pH7.8, 2 mM EDTA, 1 mM DTT, 10% glycerol, 1% Triton X-100). Total protein extract was obtained by four freezing-thawing cycles in liquid nitrogen and centrifugation at 4°C at 13,000 g. Protein expression was detected as per standard Western blot procedures. Primary antibodies used were as follows: mouse anti-β-Gal (1/2000, Sigma), rabbit anti-GFP (1/5000, Molecular Probes); and rabbit anti-HRP (1/5000, Jackson ImmunoResearch) were used as secondary antibodies. Each experiment was performed in triplicate.

β-Gal assays were performed to measure the levels of LacZ for each replicate. 20 μL of protein extract mixed with 50 μL of β-Gal assay buffer (60 mM $Na_2PO_4$, 40 mM $NaH_2PO_4$, 10 mM KCl, 1 mM $MgCl_2$, and 50 mM β-mercaptoethanol) containing ONPG was incubated at 37℃. The reaction was stopped by adding 50 μL of 1M $Na_2CO_3$ once the solution turned yellow, DO was analyzed at 415 nm. The levels of GFP were normalized to the LacZ value in each blot and were quantified using ImageJ software. The background was subtracted from each band value and the average was calculated.

Same protocol was used for *unc5* S2 cell co-transfection assays.

## Reverse transcription and qRT-PCR

Total RNA was extracted from S2 cells using tri-reagent, 1 μg of purified RNA was reverse transcribed by SuperScript II. qPCR was performed with the Roche LightCycler 480 and Sybr Green Master mix (Roche) using the following oligonucleotides:

GFP forward: ACATGAAGCAGCACGACTTCT
GFP reverse: TTCAGCTCGATGCGGTTCA
Gcm WT forward: 5′GAGAGATCTTATCCCGATCCCCTAGC3′
Gcm WT reverse: 5′CTACTACTACAGCAATACGGG3′
LacZ forward: TGTGCCGAAATGGTCCATCA
LacZ reverse: GTATCGCCAAAATCACCGCC

For each gene, the expression levels were automatically calculated (LightCycler480 Software, release 1.5.0) by calibration to gene-specific standard curves generated on input cDNAs. Collected values, derived from three amplification reactions, each performed in three independent experiments, were normalized to β-gal mRNA amounts.

## Immunolabeling and antibodies

Pupae of desired stage were collected and fixed in 4% PFA PBS (paraformaldehyde in phosphate buffer saline) overnight at 4℃. They were dissected in PBT (PBS Triton-X100, 0.3%) and wings were given four quick washes of 10 min in PBT and were incubated in the blocking reagent PBT-NGS (5% normal goat serum in PBT) for 60 min at room temperature on a planar shaker. Samples were then incubated overnight in primary antibodies (diluted in PBT-NGS): mouse-anti-Repo labels glia (1:800) and mouse-anti-22c10 labels neurons (1:1000) (DSHB), rabbit-anti-HRP labels neurons (1:1000) (DSHB), chicken-anti-GFP (1:1000) (Abcam), rat anti-Elav labels neurons (1:1000) (DSHB), rabbit-anti-Unc5 and rabbit-anti-Fra (1:500) were gifts from Benjamin Altenhein. After four washes in PBT, wings were incubated for 2 hr at room temperature in secondary antibodies (1:500) raised in mouse, rat, rabbit or chicken and coupled to Cy3, Cy5 or FITC fluorescent dyes diluted in PBT-NGS. Following a final wash in PBT, wings were mounted on slides in Aqua- Poly/Mount medium (Polysciences Inc.).

## In vivo imaging

Time-lapse analyses were performed using the standard procedure as described by *Aigouy et al. (2004, 2008)*, *Kumar et al. (2015)*, and *Soustelle et al. (2008)*. Photo bleaching was avoided by using low magnification and reduced exposure time. Maximum projections for time-lapse and confocal images were obtained by using the ImageJ software. Images were annotated by using Adobe Photoshop and Illustrator.

Quantification methods were defined previously (*Berzsenyi et al., 2011*; *Kumar et al., 2015*). Briefly, based on the time-lapse movies, the migratory process was subdivided in three phases: the earliest one describes migration initiation around 18 hAPF; this time point defines the movement of the soma of the first cell at the front of the chain. The intermediate phase identifies the time at which the glial chain reaches the level of the nerve on the costa, which is around 22 hAPF; the latest phase refers to migration completion, upon connection of the chain with the proximal glia located on the radius nerve, at around 28 hAPF, see panels shown in *Figure 1a–c*. The time points (hAPF) that were used to define these three migratory phases were calculated on the basis of time-lapse movies performed in at least 10 control animals (n≥10). In order to quantify the migratory defects, the differences in time between the control and mutant wings were compared and the graphs were plotted using the standard Student's *t* test method.

## Statistical analysis

The number of wings dissected for each experiment and genotype were more than or equal to 30. The Migratory Index (*Kumar et al., 2015*) defines the percentage of wings in which glial cells have completed migration at a given time point (28 hAPF in most cases). Graphs were made using Prism software, and the Student's *t* test method was used for the comparison between two different experimental sets. Error bars indicate the standard error mean (s.e.m.). p values: ***p<0.0001; **p <0.001; *p<0.05.

## Acknowledgements

We thank Benjamin Altenhein, Thomas Kidd, Barry Dickson, Matthias Landgraf, Iris Salecker, Wesley B Grueber, DHSB, VDRC, Kyoto Stock Center and the Bloomington Stock Center for reagents and flies. We thank Claude Delaporte, Celine Diebold and the fly, cell separation and imaging facilities for technical assistance. We thank Yoshi Yuasa, Wael Bazzi as well as the other members of the lab for valuable input and comments on the manuscript. INSERM, CNRS, UDS, Hôpital de Strasbourg, ARC, INCA, Indo-French Center for the Promotion of Advanced Research (CEFIPRA), Fondation pour la Recherche Médicale en France (FRM) and ANR grants supported this work. T Gupta, A Kumar were funded by AFM, ARC, CEFIPRA and FRM fellowships, respectively, and PB Cattenoz is supported by the ANR. The IGBMC was also supported by a French state fund through the ANR labex.

## Additional information

### Competing interests

KV: Senior editor, *eLife*. The other authors declare that no competing interests exist.

### Funding

| Funder | Grant reference number | Author |
|---|---|---|
| CEFIPRA-4403-1 | graduate student fellowship | K VijayRaghavan<br>Angela Giangrande |
| Agence Nationale de la Recherche | international award | Angela Giangrande |
| Fondation pour la Recherche Médicale | labelisation | Angela Giangrande |
| ARC Centre of Excellence for Coherent X-Ray Science | Projet grant | Angela Giangrande |
| Ligue Contre le Cancer | Grant regional | Angela Giangrande |
| USIAS | | Angela Giangrande |

The funders had no role in study design, data collection and interpretation, or the decision to submit the work for publication.

### Author contributions

TG, Conception and design, Acquisition of data, Analysis and interpretation of data, Drafting or revising the article, Contributed unpublished essential data or reagents; AK, Conception and design, Acquisition of data, Contributed unpublished essential data or reagents; PBC, Helped finalizing the revised version of the manuscript, Acquisition of data, Analysis and interpretation of data, Drafting or revising the article; KV, Hepled analyzing the adat anad suggested several experiments for the project, Analysis and interpretation of data; AG, Conception and design, Analysis and interpretation of data, Drafting or revising the article

### Author ORCIDs

K VijayRaghavan, http://orcid.org/0000-0002-4705-5629
Angela Giangrande, http://orcid.org/0000-0001-6278-5120

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
