## [Decision Letter]

Thank you for submitting your article "The Glide/Gcm fate determinant controls initiation of collective cell migration by regulating Frazzled" for consideration by *eLife*. Your article has been reviewed by three reviewers who are really familiar with the topic. They all three are positive about your work and we would be glad to consider a revised manuscript for publication in *eLife*. The evaluation has been overseen by a Reviewing Editor (Hugo Bellen) and a Senior Editor.

The reviewers have discussed the reviews with one another and the Reviewing Editor has drafted this decision to help you prepare a revised submission.

Summary:

Gupta et al. investigate here the function of netrins and their frazzled (Fra) and Unc5 receptors in glial cell chain migration in the *Drosophila* wing. Their findings show that frazzled is the major Netrin receptor facilitating initiation of glial migration, and that an early glial fate determinant TF Gcm directly activates frazzled expression. Furthermore, Netrin B but not Netrin A, apparently forms a long-range gradient that guides glia in their migration toward the CNS along peripheral nerves., and Unc5 serves as a repulsive receptor for migration of these wing glia. Transcriptional regulation of axon guidance cues and their receptors remains to be studied in detail. Therefore, this manuscript provides evidence extending our understanding of how these cues and receptors are utilized by glia and how they are transcriptionally regulated. In general, this paper appears to be one that could be of interest to the field and appropriate for *eLife* following revisions that address the concerns raised below.

1) The lab has previously demonstrated that Pioneer cells trigger chain migration. It would greatly improve this initial part of the paper, if the data on expression of Frazzled on migrating glial cells would be shown more convincingly. Instead of showing one confocal image in Figure 1 show nerves of wild type, and nerves of animals with glial or neuronal knockdown of Frazzled. There is a MiMIC insertion line available which would allow easy imaging of Frazzled protein expression (Nagarkar-Jaiswal et al., *eLife*, 2015).

Fra and Unc5 staining is not easy to see and compare across different conditions; in Figure 2, Figure 5, Figure 7, etc. GFP does not seem to match between different panels, such as in Figure 5 compared to C', or Figure 5 compared to D'. The white used here makes it difficult to separate individual RGB colors when these images are analyzed. It would be far easier for the reader to appreciate these results if there were separate panels for each staining in these figures, especially for Fra and UNC5 ICC. It would be even better to quantify the change in Fra staining in glia vs. neurons in the Fra-KD and GOF experiments using confocal optical sections.

2) It would be quite interesting and relevant in the light that collective migration is triggered by a few pioneers if a clonal analysis could be performed using the existing frazzled null allele. Do all cells have to express Fra? Provided that the wild type twin spot is allowed to stay, do these glial cells overtake mutant cell in front of them?

3) The observed genetic interaction between GCM and FRA is interesting. The fact that the *fra^[3]^* phenotype is much worse in the background of a heterozygous rA87 insertion – which is a viable enhancer trap in the *gcm* promoter region and thus a very weak hypomorph – makes me wonder how the *fra^[3]^* phenotype would look in trans to a *gcm* null allele? Is it possible to quantify the reduction of *gcm* mRNA in the different genetic backgrounds?

The authors use a migratory index to characterize the different genotypes. This is clear and straightforward but I wonder given the dramatic differences between 28APF and 24APF whether general developmental problems caused by the expression of the different transgenes also in many cells outside the nervous system (e.g. *gcm* is active also in the hemocyte lineages?) contribute to the phenotype. This could be controlled by for example comparing the time it takes to form the puparium and to eclose from the pupal cases.

4) Fra is also expressed at high levels (Figure 1) in the axons that glia migrate along, raising the question of whether neuronal Fra influences glia migration, either directly or indirectly. What happens if Fra is knocked down specifically in neurons?

The authors claim that changes in Gcm expression lead to changes of Fra in glia. However, changes of Gcm expression in glia (using *gcm*-GAL4) seem to have profound effects on neuronal Fra expression. For example, in Figure 5, knockdown of Gcm using *gcm*-GAL4 leads to a decrease of Fra in L1, and even in L3 which is not wrapped by GFP+ processes in these images. Again, in Fra GOF (Figure 5), the Fra signal co-localizes more with red 22C10 staining than with green GFP. Is this due to leaky expression of *gcm*-GAL4 in neurons at earlier stages? Or, does this suggest non-cell autonomous functions of glial Gcm on neuronal Fra expression?

5) Pioneer glia vs. follower glia: during glia chain migration, pioneer glia at the tip of the chain are suggested to actively lead followers. Fra, however, appears to be detected all along the chain (Figure 1 and Figure 1—figure supplement 1), raising the question as to whether follower glia also use Fra to initiate and direct their migration independent of pioneer glia. It is technically challenging to use GAL4 drivers to differentially target pioneer or follower glia. However, have the authors considered using MARCM (Aigouy et al., 2008) to randomly target different glia in the wing and monitor their migratory behavior in either WT or Fra mutant clones? This could also address the cell autonomous vs cell-type autonomous (RNAi experiment) requirement for Fra in wing glia migration.

6) It is surprising that the NetB mutant (NetBΔ/Y, Figure 6) has migration defects of similar severity to the Fra/+ heterozygote, and has milder effects than many other conditions, such as *gcm>fra*-KD and *gcm>gcm*-KD. This could be explained by NetB signaling through Unc5 to inhibit glial migration in Fra LOF. If this is true, the NetB mutant should enhance the Fra/+ heterozygote phenotype. Is this the case? Also, does ectopic expression of NetB in a different wing sub-region using drivers such as en-GAL4 or ap-Gal4 mis-direct glial migration?

7) In the subsection “*fra* plays an instructive role in L1 glia migration”, the authors claim 'Finally, we measured the migration speed at the time of initiation and it was not higher in Fra GOF glia (10 μm/h) than in control glia (40 μm/h)'. This is confusing since the numbers do appear to be significantly different. What is the actual speed of glial migration in Fra GOF vs. control? The authors should also include this important information in graphs using either the speed or duration of glial migration from initiation to completion of migration.

8) The authors try to rescue RNAi phenotypes by co-expression of RNAi and UAS-GOF constructs, stating that this 'excludes the possibility of off target effects'. This is not accurate since this rescue effect could be due to titration of siRNA away from the endogenous mRNA by overexpression of the exogenous mRNA. To directly rule out off target effects the rescue experiments should be done by using either a UAS construct bearing mutants that are insensitive to RNAi or perhaps an RNAi line that targets non coding regions.

9) Throughout the study, there are no details or explanations on the methods of quantification. For instance, Results: subsection “*fra* plays an instructive role in L1 glia migration”, third paragraph – this is very unclear! how exactly the authors define "initiation? and how was the quantification done? Figure 2, how does author calculate the hAPF for initiation, costa reach, and complete migration? what is the n number? how early or late in migration do the authors consider a defect in initiation? These all have to be explained fully in the text (the Methods section or the legends). Without these knowledge, the readers will have a hard time to follow and do not understand how the conclusion was reached. Also, for the time-lapse figure, it is not clear what does 22:48 hAPF mean since there are different genotypes shown, and I cannot find any information on this in the legends nor methods.

10) Panel organization: In Figure 2, Figure 3, Figure 6, Figure 7, in which the authors characterize glia migration phenotypes (MI), genotypes are assigned in different panels with the same or different time points. These arrangements make it difficult to compare different genotypes. For example, Figure 2 could be combined, and the same is true for Figure 3 and Figure 7 and F. In some panels, more genotypes should be included. For example, *gcm*-GAL4>*gcm* (GOF) is needed in Figure 3, and NetB-GAL4> NetB needs to be added to Figure 6. In Figure 6, all three genotypes should be compared in both time points.

11) Since Gcm is a very important factor in controlling neuronal and glial cell fates, it is crucial to address clearly whether this conversion function is in any part related to the migration phenotype the authors are trying to assessing in the study. However, there is only one experiment provided which wasn't able to give a clear answer. For instance, Figure 4—figure supplement 1: it is not common to analyze the co-positive cells of *gcm*>GFP and Elav as Elav stained for nucleus and GFP labels the membrane. Can't the author do the costaining of Repo and Elav? this might be more clear and definitive. Also, what is the percentage of *gcm* KD wings not having extra Elav positive cells? Is it high or low? Isn't it plausible that a high percentage of the *gcm* KD wings should have extra Elav cells? The *gcm* KD wings with normal Elav positive cells might as well mean that *gcm* expression wasn't knock-down and in that case why is the MI still defective? How do the authors reconcile the discrepancies? This section has to be explained clearly since glia-neuron switch is a very important function for Gcm and whether the migration function depends on this needs to be clearly addressed with clear, precise experimental results.

[Editors' note: further revisions were requested prior to acceptance, as described below.]

Thank you for resubmitting your work entitled "The Glide/Gcm fate determinant controls initiation of collective cell migration by regulating Frazzled" for further consideration at *eLife*. Your revised article has been evaluated by a Senior editor, a Reviewing editor, Hugo Bellen, and three reviewers.

The manuscript has been improved but there are some remaining issues that need to be addressed. The reviewers discussed their reviews on line and came to a consensus that they all agreed with reviewer 3. The reviewers feel that some of the points that they raised where not addressed and Reviewer #3 summarized these concerns.

*Reviewer #3:*

I read the revised paper and cannot support publication at this stage due to the poor demonstration that *fra* is expressed by the glia and the lack of a clear demonstration that the phenotype is due to loss of glial *fra* function. I admit there are some hints supporting this suggestion but I do not think the data are conclusive.

Expression of *fra* in glia. The authors added a new Figure 1 which shows *fra* expression upon glial knockdown. For me this is *not* showing that *fra* is expressed in glia. There is as much background as in the wild type and the control experiment, neuronal knockdown, is not shown based on the argument that elav is also expressed by glia. Please, use nsybGal4 which is specific – and/or generate clones! The quality of the antibody stainings is not consistent and for example in Figure 2 glial knockdown is also removing the neuronal Fra expression. For me the quality of the data is still simply not good enough.

The phenotypes are based on somewhat odd genetic backgrounds. May be there is a *fra* haploinsufficiency and heterozygous *fra*/+ animals show a phenotype. But to really determine the Fra phenotype, a true loss of function analysis is required. The argument that MARCM clones are hard to get is weird as the same lab has generated these clones before in another study. Glia-specific *fra* knockdown animals show roughly the same phenotype as heterozygous *fra3*/+ animals (MI 58 vs.61). Still only the RNAi induced phenotype can be completely rescued. The authors conclude that this may be due to the reduction of neuronal Fra expression (which was not tested). However, one could also argue that the authors are looking at a genetic background effect. Thus, a MARCM analysis is mandatory.

We had asked for the effect of *gcm* on the migratory phenotype of *fra3*/+ heterozygous animals. Although the authors provide arguments for an allelic series of the different *gcm* alleles they do not show the effect of a true *gcm* loss of function allele on the migratory index. Provided that the presence of one very weak, homozygous viable, *gcm* allele has such a profound effect on the migratory index, what would a true null allele do? The allelic series shown in the rebuttal letter indicates that homozygous *gcm*>GFP animals have the same effect as *gcm*>GFP gcmN7-4. Since *gcm*N7-4 is a null this implies that *gcm*>GFP is a null as well – but why do they survive then? Something is odd here.

---

## [Author Response]

[…]

*1) The lab has previously demonstrated that Pioneer cells trigger chain migration. It would greatly improve this initial part of the paper, if the data on expression of Frazzled on migrating glial cells would be shown more convincingly. Instead of showing one confocal image in Figure 1 show nerves of wild type, and nerves of animals with glial or neuronal knockdown of Frazzled. There is a MiMIC insertion line available which would allow easy imaging of Frazzled protein expression (Nagarkar-Jaiswal et al., eLife, 2015).*

*Fra and Unc5 staining is not easy to see and compare across different conditions; in Figure 2, Figure 5, Figure 7, etc. GFP does not seem to match between different panels, such as in Figure 5 compared to C', or Figure 5 compared to D'. The white used here makes it difficult to separate individual RGB colors when these images are analyzed. It would be far easier for the reader to appreciate these results if there were separate panels for each staining in these figures, especially for Fra and UNC5 ICC. It would be even better to quantify the change in Fra staining in glia vs. neurons in the Fra-KD and GOF experiments using confocal optical sections.*

As suggested by the referees we made sure that the labeling was indeed less prominent in the KD and more intense in the overexpressing animals. We have now included the data that also shows the glial knockdown of *fra* in a 18 hAPF old wing in Figure 1 and we have included panels that only shows Fra or Unc5 labeling in Figure 2, Figure 5 and Figure 7.

The difference in GFP intensity in the merged panels is due to the strong red signal. Specifically, at late stages these differences are due to the fact that glia wrap neurons (which are in red) completely. In addition, we have quantified Fra staining in Figure 1 between control and *fra* KD L1 glia by including heat maps. Note that in the KD the neuronal (but not the glial) Fra signal is as high if not higher than in the wt.

Concerning the MiMIC insertion, we tried and stay consistent, using the same antibody that was used in previous papers, in particular in the paper on glial migration from Altenhein and collaborators (References on the used antibody: Von Hilchen et al., 2010; Timofeeev et al., 2012; Joly et al., 2007; Kolodziej et al., 1996; Orr et al., 2014; Gong et al., 1999; Pert et al., 2015).

*2) It would be quite interesting and relevant in the light that collective migration is triggered by a few pioneers if a clonal analysis could be performed using the existing frazzled null allele. Do all cells have to express Fra? Provided that the wild type twin spot is allowed to stay, do these glial cells overtake mutant cell in front of them?*

We see Fra expression in all L1 glia, which is shown in Figure 1—figure supplement 1(subsection “Frazzled expression in the glia of the developing Drosophila wing”, second paragraph). The clonal analysis is a valuable point, however we think that the data from this experiment will not be easy to interpret. Fra is also expressed in neurons, therefore getting clones that only affect glia will be difficult (low division rate) and even more to get clones that specifically affect the pioneer cells. We did in the past get MARCM clones in pioneer cells, but this is a very rare event. This was useful to analyze the morphology of pioneers, but the analysis of migration phenotypes requires high number of events in order to perform the statistics. More importantly, we have previously shown that several pioneers are at work, therefore mutating one or two of them may not be sufficient to see a phenotype.

*3) The observed genetic interaction between GCM and FRA is interesting. The fact that the fra^3^ phenotype is much worse in the background of a heterozygous rA87 insertion – which is a viable enhancer trap in the gcm promoter region and thus a very weak hypomorph – makes me wonder how the fra^3^ phenotype would look in trans to a gcm null allele? Is it possible to quantify the reduction of gcm mRNA in the different genetic backgrounds?*

*The authors use a migratory index to characterize the different genotypes. This is clear and straightforward but I wonder given the dramatic differences between 28APF and 24APF whether general developmental problems caused by the expression of the different transgenes also in many cells outside the nervous system (e.g. gcm is active also in the hemocyte lineages?) contribute to the phenotype. This could be controlled by for example comparing the time it takes to form the puparium and to eclose from the pupal cases.*

Concerning the interaction, *gcm^rA87^*is a weaker allele than the *gcm‐Gal4* line and, accordingly *gcm^rA87^/fra^3^*has a weaker migratory phenotype that *gcm‐Gal4/ fra^3^*. These data are now shown together in Figure 3, to comply with reviewers’ suggestion. For the benefit of the reviewers, we also show here below the rate of conversion and/or the number of repo positive cells in different allelic combinations (*gcm-Gal4/ gcm^rA87^, gcm-Gal4/ gcm-Gal4, gcm^rA87^/ gcmN7-4* and *gcm‐Gal4>gcm RNAi*). The inverse correlation between the two phenotypes in the different backgrounds further sustains our hypothesis. We believe that the phenotypic analysis of the allelic series is more sensitive than RNA quantification, given the low number of *gcm* expressing cells. For the sake of simplicity, we do not think that the data here below should be included. Should the referee feel it necessary, however, we are eager to include them in Figure 5—figure supplement 1.

We did not observe any general developmental defect, such as delay in eclosion from the pupal case.

Author response image 1.**DOI:**
http://dx.doi.org/10.7554/eLife.15983.028

To rule out a possible hemocyte contribution in glia migration we downregulated *gcm* using a hemocyte driver collagen‐*Gal4* and found no effect on glia migration. Further, we also knocked down *fra* using a temperature sensitive *Gal80* construct of *gcm* driver and specifically knocked down *fra* post embryonically. Since *gcm* is only expressed in blood cells embryonically, this eliminates any possible contribution of hemocyte Gcm expression to the observed phenotypes. These analyses confirm that *gcm* and *fra* directly affect glia migration as in these conditions too glia migration is significantly delayed (Figure 5—figure supplement 1).

*4) Fra is also expressed at a high levels (Figure 1) in the axons that glia migrate along, raising the question of whether neuronal Fra influences glia migration, either directly or indirectly. What happens if Fra is knocked down specifically in neurons?*

*The authors claim that changes in Gcm expression lead to changes of Fra in glia. However, changes of Gcm expression in glia (using gcm-GAL4) seem to have profound effects on neuronal Fra expression. For example, in Figure 5'-D', knockdown of Gcm using gcm-GAL4 leads to a decrease of Fra in L1, and even in L3 which is not wrapped by GFP+ processes in these images. Again, in Fra GOF (Figure 5), the Fra signal co-localizes more with red 22C10 staining than with green GFP. Is this due to leaky expression of gcm-GAL4 in neurons at earlier stages? Or, does this suggest non-cell autonomous functions of glial Gcm on neuronal Fra expression?*

The neuronal expression of Fra is an important issue and we thank the reviewer(s) for the comment. We analyzed L1 axons of 17 hAPF *fra^3^*wings and found that about 30% of them show a transient phenotype of delayed axonal growth (see now Figure 3—figure supplement 2). We also analyzed wings where *fra* is knocked down using an Elav driver and got a very strong migratory phenotype (Data not shown in the manuscript). However, the available Elav drivers (we tested two of them) are also expressed in wing glia and epithelial cells (Figure 11). Other known drivers are either expressed in neuronal subsets (motoneurons) or too late for our purpose. Thus, in the wing, we were unable to use a driver that specifically activates or knocks down gene expression in neurons. These analyses suggested that downregulating *fra* in neurons may indirectly affect glia migration and prompted us to use a driver that specifically affects glia.

Author response image 2.25 hAPF immunolabeled wing of the *elav-Gal4,UASmCD8GFP* transgenic line (anti-GFP in green and anti-Repo in red).Note the presence of the mCD8GFP in the axons, in the glia (Repo-positive cells in red, arrows) and in the epithelium (asterisk).**DOI:**
http://dx.doi.org/10.7554/eLife.15983.029

We think there is no change in the neuronal expression of Fra upon *gcm* KD or GOF. Figure 5 is only one example and the decrease might be due to the fact that these images are few sections and not the whole projection. Concerning the role of Gcm on neuronal expression of Fra. We never observed the Gcm driver being expressed in neurons. For all the above reasons, we do not think there is a non-cell autonomous effect.

*5) Pioneer glia vs. follower glia: during glia chain migration, pioneer glia at the tip of the chain are suggested to actively lead followers. Fra, however, appears to be detected all along the chain (Figure 1 and Figure 1—figure supplement 1), raising the question as to whether follower glia also use Fra to initiate and direct their migration independent of pioneer glia. It is technically challenging to use GAL4 drivers to differentially target pioneer or follower glia. However, have the authors considered using MARCM (Aigouy et al., 2008) to randomly target different glia in the wing and monitor their migratory behavior in either WT or Fra mutant clones? This could also address the cell autonomous vs cell-type autonomous (RNAi experiment) requirement for Fra in wing glia migration.*

The reviewers raise an interesting point, i.e. what is the role of Fra, since it is all along the L1 chain. We believe that the suggested MARCM clonal analysis is beyond the scope of this manuscript. We indeed used MARCM to follow the morphology of single pioneers but mutant clones on single pioneer glial cells are very rare, which makes it hard to draw conclusions that are statistically significant. If we go for big clones, they will also affect neurons, which will not help assessing the cell autonomous effect. See also answer to Q2.

To comply with the comment, we asked whether the ligand Net B accumulates on specific glia, based on the recent paper from Pert et al., (Biology Open 2015) showing Net B accumulation in Fra expressing cells. Immunolabeling assays on young wings indeed show preferential Net B accumulation in some cells at the front of the chain. We speculate that the receptor is all over the glia but pioneer glia are closer to the ligand than the other glia and some pioneers accumulate it, perhaps due to receptor‐mediated endocytosis. This may in turn triggers the ‘promigratory’ downstream pathway in those cells.

Author response image 3.**DOI:**
http://dx.doi.org/10.7554/eLife.15983.030

*6) It is surprising that the NetB mutant (NetBΔ/Y, Figure 6) has migration defects of similar severity to the Fra/+ heterozygote, and has milder effects than many other conditions, such as gcm>fra-KD and gcm>gcm-KD. This could be explained by NetB signaling through Unc5 to inhibit glial migration in Fra LOF. If this is true, the NetB mutant should enhance the Fra/+ heterozygote phenotype. Is this the case? Also, does ectopic expression of NetB in a different wing sub-region using drivers such as en-GAL4 or ap-Gal4 mis-direct glial migration?*

We thank the reviewer(s) for the thoughtful comments. The reviewer is correct, the NetB phenotype might be milder than that observed in *fra3/+* because of the effects of NetB on *unc5*. Along this line, Figure 8—figure supplement 1 shows that *UAS‐RNAi* for both *unc5* and *fra* has a weaker phenotype than *UAS‐fra‐RNAi* alone. As per the double mutant *NetBΔ/Y; fra^3^/+*, this genotype would not inform on the role of Unc5, because in a hemizygous *NetB*Δ*/Y* animal both Fra and Unc5 should be absent anyway, therefore removing the *fra* receptor would not make any difference. We thought that this experiment may nevertheless be informative and tell whether *fra* is also submitted to other signals, as suggested by recent studies (Neuhaus-Follini et al; 2015). We hence performed the suggested experiment and it does seem that *fra* also depends on other signals, see data on the right graph. We would not include this data in the main text, however, as they go beyond the scope of the manuscript, which is to show the impact of a fate determinant on late aspects of cell specification. As requested, we show the data on the ectopic expression of NetB using *en‐Gal4* in the posterior wing compartment and using *GMR 29F05-Gal4* that is expressed in the distal part of the wing. These experiments show that ectopic NetB does not affect glia migration (Now Figure 8—figure supplement 1). Thus, NetB does not act as a guidance molecule triggering directionality. This is also discussed in the manuscript on page 23.

Author response image 4.**DOI:**
http://dx.doi.org/10.7554/eLife.15983.031

*7) In the subsection “fra plays an instructive role in L1 glia migration”, the authors claim 'Finally, we measured the migration speed at the time of initiation and it was not higher in Fra GOF glia (10 μm/h) than in control glia (40 μm/h)'. This is confusing since the numbers do appear to be significantly different. What is the actual speed of glial migration in Fra GOF vs. control? The authors should also include this important information in graphs using either the speed or duration of glial migration from initiation to completion of migration.*

To comply with the comments and clarify our statements, we performed further analyses that are now incorporated in Figure 3—figure supplement 2. The time-lapse analysis shows that *fra* GOF glia start moving earlier than control glia (Figure 2, Figure 3—figure supplement 1). The *gcm*>*fra* GOF glia phenotype is solely due to altered initiation rather than to acceleration and we have added a speed graph of control and *fra* GOF glia in Figure 3—figure supplement 2. The speed analysis was done on the most proximal glial cell of the chain (μm/h, y-axis; hAPF, x‐axis). The distance covered by the front cell was measured by analyzing the position of the glial soma every hour, as already shown in Kumar et al., 2015. We followed the cell starting from the point of initiation till complete migration. The migratory speed increases both in control and *fra* GOF glia until they reach the level of costa and then it decreases, however, *fra* GOF glia display a more uniform speed than that of control glia: they accelerate less in the early phases and slow down less in the late phases.

*8) The authors try to rescue RNAi phenotypes by co-expression of RNAi and UAS-GOF constructs, stating that this 'excludes the possibility of off target effects'. This is not accurate since this rescue effect could be due to titration of siRNA away from the endogenous mRNA by overexpression of the exogenous mRNA. To directly rule out off target effects the rescue experiments should be done by using either a UAS construct bearing mutants that are insensitive to RNAi or perhaps an RNAi line that targets non coding regions.*

We take the point made by the reviewer. However, we would like to emphasize that the *RNAi* data are further supported by the opposite phenotype obtained in gain of function experiments and by the fact that we could rescue the *fra^3^*mutant phenotype by reintroducing a UAS‐*fra* transgene in glia (Figure 3—figure supplement 1). We nevertheless played down the interpretation of the RNAi data in the main text, to comply with the comment.

*9) Throughout the study, there are no details or explanations on the methods of quantification. For instance, Results: subsection “fra plays an instructive role in L1 glia migration”, third paragraph – this is very unclear! how exactly the authors define "initiation? and how was the quantification done? Figure 2, how does author calculate the hAPF for initiation, costa reach, and complete migration? what is the n number? how early or late in migration do the authors consider a defect in initiation? These all have to be explained fully in the text (the Methods section or the legends). Without these knowledge, the readers will have a hard time to follow and do not understand how the conclusion was reached. Also, for the time-lapse figure, it is not clear what does 22:48 hAPF mean since there are different genotypes shown, and I cannot find any information on this in the legends nor methods.*

As the referee suggests, we included a detailed description of how the time-laps analysis was performed and how the quantification was made in the Methods section of the manuscript (in vivoimaging paragraph). These methods were already defined in our previous publications (Berzseny et al., 2011, Kumar et al., 2015).

“Based on the time-lapse videos, the migratory process has been subdivided in three phases: the earliest one describes migration initiation around 18 hAPF; this time point defines the movement of the soma of the first cell at the front of the chain. […] In order to quantify the migratory defects, the differences in time between the control and mutant wings were compared and the graphs were plotted using the standard Student's t test method.”

We have used the above phases to select the images from the time-lapsevideos. Typically, Figure 2shows a snap shot at 21h48, which corresponds to the time by which wild type glia have reached the Costa, in average. Figure 3,shows a snap shot at 22h48, which corresponds to the time by which GOF glia have completed migration, in average. We have clarified this in the text.

*10) Panel organization: In Figure 2, Figure 3, Figure 6, Figure 7, in which the authors characterize glia migration phenotypes (MI), genotypes are assigned in different panels with the same or different time points. These arrangements make it difficult to compare different genotypes. For example, Figure 2 could be combined, and the same is true for Figure 3 and Figure 7 and F. In some panels, more genotypes should be included. For example, gcm-GAL4>gcm (GOF) is needed in Figure 3, and NetB-GAL4> NetB needs to be added to Figure 6. In Figure 6, all three genotypes should be compared in both time points.*

As per referee’s suggestion we have reorganized Figure 2; Figure 3 and Figure 7into one panel. We have also included the migratory index of *gcm>gcm* GOF in Figure 3and the migratory index of *NetB>NetB* GOF in Figure 6. Finally, as requested, we have also included all the genotypes in Figure 6 and G.

*11) Since Gcm is a very important factor in controlling neuronal and glial cell fates, it is crucial to address clearly whether this conversion function is in any part related to the migration phenotype the authors are trying to assessing in the study. However, there is only one experiment provided which wasn't able to give a clear answer. For instance, Figure 4—figure supplement 1: it is not common to analyze the co-positive cells of gcm>GFP and Elav as Elav stained for nucleus and GFP labels the membrane. Can't the author do the costaining of Repo and Elav? this might be more clear and definitive. Also, what is the percentage of gcm KD wings not having extra Elav positive cells? Is it high or low? Isn't it plausible that a high percentage of the gcm KD wings should have extra Elav cells? The gcm KD wings with normal Elav positive cells might as well mean that gcm expression wasn't knock-down and in that case why is the MI still defective? How do the authors reconcile the discrepancies? This section has to be explained clearly since glia-neuron switch is a very important function for Gcm and whether the migration function depends on this needs to be clearly addressed with clear, precise experimental results.*

Author response image 5.**DOI:**
http://dx.doi.org/10.7554/eLife.15983.032

To address this issue we included the co-‐labeling of Repo and Elav in (Figure 5—figure supplement 1), as requested. Concerning the possible indirect effects of glia to neuron conversion on migration, we find that only 14% of the *gcm* KD wings show conversion, whereas 75% of the KD wings show delayed migration. Further, to make sure that glia to neuron transformation in *gcm* knockdown wings does not affect glia migration we analyzed two sets of KD wings. The first set comprises of wings with fate transformation and the migratory index of these wings is very low, 25%. The second set includes wings without fate transformation; in this case, the migratory index is 45%, which is still significantly different from the control (Figure 5—figure supplement 1). Last but not the least, the number of converted cells is very low (10). We also analyzed an allelic combination that is weaker than the KD and that only shows fate conversion in 10% of the wings. We still find a migratory delay in these wings (*gcm‐Gal4/gcm^rA87^*animals, See Figure 3). This analysis confirms that Gcm affects glia migration irrespective of the change in the number of glial cells due to the fate transformation, which altogether fits perfectly with the finding that Gcm directly induces the expression of the Fra receptor.

[Editors' note: further revisions were requested prior to acceptance, as described below.]

[…]

*Reviewer #3:*

*I read the revised paper and cannot support publication at this stage due to the poor demonstration that fra is expressed by the glia and the lack of a clear demonstration that the phenotype is due to loss of glial fra function. I admit there are some hints supporting this suggestion but I do not think the data are conclusive.*

Concerning the detailed points described in, we addressed the three issues, as previously agreed:

1) Better evidence for Fra expression/requirement in glia

2) Migratory assay in *fra4/gcm null* wings

3) Mitotic clones

*Expression of fra in glia. The authors added a new Figure 1which shows fra expression upon glial knockdown. For me this is not showing that fra is expressed in glia. There is as much background as in the wild type and the control experiment, neuronal knockdown, is not shown based on the argument that elav is also expressed by glia. Please, use nsybGal4 which is specific – and/or generate clones! The quality of the antibody stainings is not consistent and for example in Figure 2 glial knockdown is also removing the neuronal Fra expression. For me the quality of the data is still simply not good enough.*

We followed the suggestion and used the nsybGal4 line as a neuronal driver. We first assessed expression of this driver in wing neurons, next we used the nsybGal4 line to knock down Fra in neurons and analyze glial migration. We found no defects in the migratory index, except in wings that carry an axonal navigation defect. These data are now presented in Figure 3—figure supplement 1. The wings carrying axon navigation defects were of course eliminated from the graph, as it has already been published that axonal defects result in glial migration defects.

We add Fra immunolabelling at 18 hAPF in combination with a second glial reporter (Repo>GFP). See Figure 1.

We removed panels H-J from Figure 1, as requested.

We improved the quality of Figure 2 showing glial Fra expression at later stages, as requested.

*The phenotypes are based on somewhat odd genetic backgrounds. May be there is a fra haploinsufficiency and heterozygous fra/+ animals show a phenotype. But to really determine the Fra phenotype, a true loss of function analysis is required. The argument that MARCM clones are hard to get is weird as the same lab has generated these clones before in another study. Glia-specific fra knockdown animals show roughly the same phenotype as heterozygous fra3/+ animals (MI 58 vs.61). Still only the RNAi induced phenotype can be completely rescued. The authors conclude that this may be due to the reduction of neuronal Fra expression (which was not tested). However, one could also argue that the authors are looking at a genetic background effect. Thus, a MARCM analysis is mandatory.*

We complied with the request of the referee and analyzed the migratory phenotype in transheterozygous animals *fra3/gcm null (N7-4*), which further confirms the data obtained with the hypomorphs and the KD line. These data are now presented in Figure 3—figure supplement 2, second last column.

We would like to clarify that the allelic series showing the reduction in the number of Repo+ cells in the last rebuttal letter shows the effects of *gcm^rA87^/gcm^N7-4^*and not *gcm*>GFP/ *gcm^N7-4^,* as claimed by the referee (see the same figure panel). We have already mentioned in the manuscript that *gcm^rA87^*is a weak *gcm* hypomorphic allele. Therefore, the effect it produces on the number of repo cells when combined with null *gcm^N7-4^*allele is similar to *gcm>*GFP/*gcm>*GFP.

*We had asked for the effect of gcm on the migratory phenotype of fra3/+ heterozygous animals. Although the authors provide arguments for an allelic series of the different gcm alleles they do not show the effect of a true gcm loss of function allele on the migratory index. Provided that the presence of one very weak, homozygous viable, gcm allele has such a profound effect on the migratory index, what would a true null allele do? The allelic series shown in the rebuttal letter indicates that homozygous gcm>GFP animals have the same effect as gcm>GFP gcmN7-4. Since gcmN7-4 is a null this implies that gcm>GFP is a null as well – but why do they survive then? Something is odd here.*

We performed mitotic clone analyses. We ordered the strain to produce MARCM clones in WT or in Fra KD cells. We also ordered strains to perform a more refined clonal analysis using the recently developed coinFLP technique in a wild type and in a Fra KD background. The crosses from this second approach went faster. Given the time frame we had agreed upon and the fact that the PhD student could only come for one month, we pursued the analyses of the coinFLP clones, which was indeed very successful. We could identify wild type and mutant clones in the same wing using two membrane reporters (GFP- green cells: WT cells; RFP – red cells: Fra DK cells). In this way, we could confirm that red (Fra KD) cells contain much less Fra than green (WT) cells. Furthermore, we could specifically identify WT and Fra KD glia using the Repo marker that is detected in glial nuclei. All these data are now shown in Figure 1.